# Spatio-temporal variations of HNO$_3$ total columns from 9 years of IASI measurements - A driver study

Gaétane Ronsmans[1], Catherine Wespes[1], Daniel Hurtmans[1], Cathy Clerbaux[1,2], and
Pierre-François Coheur[1]

[1]Université Libre de Bruxelles (ULB), Faculté des Sciences, Chimie Quantique et Photophysique, Brussels, Belgium
[2]LATMOS/IPSL, UPMC Univ. Paris 06 Sorbonne Universités, UVSQ, CNRS, Paris, France

*Correspondence to:* Gaétane Ronsmans (gronsman@ulb.ac.be)

**Abstract.** This study aims at understanding the spatial and temporal variability of HNO$_3$ total columns in terms of explanatory variables. To achieve this, multiple linear regressions are used to fit satellite-derived time series of HNO$_3$ daily averaged total columns. First, an analysis of the IASI 9-year time series (2008-2016) is conducted based on various equivalent latitude bands. The strong and systematic denitrification of the southern polar stratosphere is observed very clearly. It is also possible to distinguish, within the polar vortex, three regions wich are differently affected by the denitrification. Three exceptional denitrification episodes in 2011, 2014 and 2016 are also observed in the northern hemisphere, due to unusually low arctic temperatures. The time series are then fitted by multivariate regressions to identify what variables are responsible for HNO$_3$ variability in global distributions and time series, and to quantify their respective influence. Out of an ensemble of proxies (annual cycle, solar flux, quasi-biennial oscillation, multivariate ENSO index, Arctic and Antarctic oscillations and volume of polar stratospheric clouds), only the ones defined as significant (p-value $< 0.05$) by a selection algorithm are retained for each equivalent latitude band. Overall, the regression gives a good representation of HNO$_3$ variability, with especially good results at high latitudes (60-80% of the observed variability explained by the model). The regressions show everywhere the dominance of the annual variability, which is related to specific chemistry and dynamic depending on the latitudes. We find that the polar stratospheric clouds (PSCs) also have a major influence in the polar regions, and that their inclusion in the model improves the correlation coefficients and the residuals. However, there is still a relatively large part of the HNO$_3$ variability that remains unexplained by the model, especially in the intertropical regions, where factors not included in the regression model (such as vegetation fires or lightning) may be at play.

## 1  Introduction

Nitric acid (HNO$_3$) is known to influence the ozone (O$_3$) concentrations in the polar regions, because of its role of NO$_x$ ($\equiv$ NO+NO$_2$) reservoir and its ability to form polar stratospheric clouds (PSCs) inside the vortex (e.g. Solomon (1999); Urban et al. (2009); Popp et al. (2009)). In the stratosphere, HNO$_3$ forms from the reaction between OH and NO$_2$ (produced by the reaction N$_2$O+O$^1$D) and is destroyed by its reaction with OH or its photodissocation, both of these reactions being slow in daytime and virtually non-existent during nighttime (McDonald et al., 2000; Santee et al., 2004). This leads to photochemical

lifetimes between 1 and 3 months up to 30 km altitude and around 10 days above (Austin et al., 1986), inducing similar trans-port pathways for $O_3$ and $NO_y$ (the sum of all reactive nitrogen species, i.e. including $HNO_3$) in general (Fischer et al., 1997). During the polar winter, with the arrival of low temperatures, PSCs, composed of $HNO_3$, sulphuric acid ($H_2SO_4$) and water ice ($H_2O$), form within the vortex (e.g. Voigt et al. (2000); von König et al. (2002)). They act as sites for heterogeneous reactions,

turning inactive forms of chlorine and bromine into active radicals, and leading to the depletion of $O_3$ in the polar regions (e.g. Solomon (1999); Wang and Michelangeli (2006); Harris et al. (2010); Wegner et al. (2012)). Furthermore, the formation of these PSCs, and particularly of the nitric acid trihydrates (NAT), leads to a denitrification of the stratosphere (condensation of $HNO_3$ followed by sedimentation towards the lower stratosphere), which prevents $ClONO_2$ to reform (e.g. Gobbi et al. (1991); Solomon (1999); Ronsmans et al. (2016)) and enhances further the depletion of ozone.

$HNO_3$ was measured by a variety of instruments in the last decades, amongst which the MLS (on the UARS, then the Aura satellite) provided the most complete data set. Its measurements started in 1991 and allowed extensive analyses on the seasonal and interannual variability, as well as on the vertical distribution of $HNO_3$ (Santee et al., 1999, 2004), with however a coarse horizontal resolution. Other instruments have also measured $HNO_3$ in the atmosphere, such as MIPAS (ENVISAT, Piccolo and Dudhia (2007)), ACE-FTS (SCISAT, Wang et al. (2007)) and SMR[1] (Odin, Urban et al. (2009)), but few of these data

have been used for geophysical analyses in terms of chemical and physical processes influencing $HNO_3$, mostly because of the limited horizontal sampling of these instruments.

On the other hand, $O_3$ has been extensively analysed, and numerous studies have been conducted to provide a better understand-ing of the factors influencing the stratospheric $O_3$ depletion processes and to assess the efficiency of the international treaties that were put in place to reduce its extent (e.g. Lary (1997); Solomon (1999); Morgenstern et al. (2008); Mäder et al. (2010);

Knibbe et al. (2014); Wespes et al. (2016)). Most recent studies have used multivariate regression analyses in order to identify and quantify the main contributors to the $O_3$ spatial and seasonal variations. The variables included in such regression models depend on the atmospheric layer investigated (troposphere or stratosphere) and most often include the solar cycle, the quasi-biennial oscillation (QBO), the aerosol loading and the equivalent effective stratospheric chlorine (EESC) (e.g. Wohltmann et al. (2007); Sioris et al. (2014); de Laat et al. (2015)). They often also include climate-related proxies for specific dynamical

patterns such as El Niño southern oscillation (ENSO), the North Atlantic Oscillation (NAO) or the Antarctic Oscillation (AAO) (Frossard et al., 2013; Rieder et al., 2013). In various multivariate regression studies, an iterative selection procedure is used to isolate the relevant variables for the concerned species (Steinbrecht, 2004; Mäder et al., 2007; Knibbe et al., 2014; Wespes et al., 2016, 2017).

Despite the fact that it is one of the main species influencing stratospheric $O_3$, $HNO_3$ has much less been studied in terms

of explanatory variables, in part because of the lack of global, consistent and sustained measurements. Identifying the factors driving its spatial and temporal variability could help to characterize its behaviour in the stratospheric chemistry, and hence its interactions with $O_3$.

The Infrared Atmospheric Sounding Interferometer (IASI) onboard the Metop satellites has been and still is providing global

---

[1]in the order of the instruments cited above: Microwave Limb Sounder, Michelson Interferometer for Passive Atmospheric Sounding, Atmospheric Chem-istry Experiment-Fourier Transform Spectrometer, Sub-Millimetre Radiometer

measurements of the HNO$_3$ total column, which are used here to investigate the HNO$_3$ spatial and temporal variability. The data set used (Section 2) consists of time series of HNO$_3$ columns retrieved from IASI/Metop A measurements over the period 2008-2016, twice daily and with global coverage. The unprecedented spatial and temporal sampling of the high latitudes allows for an in-depth monitoring of the atmospheric state in particular during the polar winter (Wespes et al., 2009; Ronsmans et al., 2016). We make use of equivalent latitudes in order to isolate polar air masses with specific polar vortex characteristics, and hence to better understand the role of HNO$_3$ in the polar chemistry, with regard to geophysical features such as the extent of the polar vortex and polar temperatures (Section 3). We next apply multivariate regressions to the IASI-derived HNO$_3$ time series to statistically characterize their global distributions and seasonal variability for the first time, and this at different latitudes. The global coverage and the sampling of observations also allow retrieving global patterns of the main HNO$_3$ drivers (Section 4).

## 2   IASI HNO$_3$ data

The HNO$_3$ columns used here were retrieved from the measurements of the IASI instrument onboard the Metop A satellite. IASI measures the upwelling infrared radiation from the Earth's surface and the atmosphere in the 645-2760 cm$^{-1}$ spectral range at nadir and off nadir along a broad swath (2200 km). The level 1C data set used for the retrieval consists of measurements taken twice a day (at 9:30 AM and PM, equator crossing time) at a 0.5 cm$^{-1}$ apodized spectral resolution and with a low radiometric noise (0.2 K in the HNO$_3$ atmospheric window) (Clerbaux et al., 2009; Hilton et al., 2012). The ground field of view of the instrument consists of 4 elliptical pixels (2 by 2) yielding a horizontal footprint (single pixel) that varies from 113 km$^2$ (12 km diameter) at nadir to 400 km$^2$ at the end of the swath.

To retrieve HNO$_3$ atmospheric concentrations, we use the level 1C measurements available in near-real time at Université Libre de Bruxelles (ULB) and retrieved by the Fast Optimal Retrievals on Layers for IASI (FORLI) software, which uses the optimal estimation method (Rodgers, 2000). A complete description of the FORLI method can be found in Hurtmans et al. (2012) and a summary of the retrieval parameters specific to HNO$_3$ in Ronsmans et al. (2016). The retrieval initially yields HNO$_3$ vertical profiles on 41 levels (from 0 to 40 km altitude) but with limited vertical sensitivity. The characterization of the retrieved profiles conducted by Ronsmans et al. (2016) showed indeed that the degrees of freedom for signal (DOFS) range from 0.9 to 1.2 at all latitudes. Because of this lack of vertical sensitivity, the HNO$_3$ total column is the most representative quantity for the IASI measurements and it is exploited here for the investigation of the HNO$_3$ time evolution. It is important to note, however, and as thoroughly discussed in Ronsmans et al. (2016), that the information on the HNO$_3$ profile comes mostly from the lower stratosphere (15-20 km) and that the profile is therefore mainly indicative of the stratospheric abundance. In order to compute the total column, the retrieved vertical profiles are integrated over the whole altitude range. Our previous study showed that the resulting total columns yield a mean error of 10% and a low bias (10.5%) when compared to ground-based FTIR measurements (Ronsmans et al., 2016). The data set used spans from January 2008 to December 2016 with daily median HNO$_3$ columns averaged on a $2.5° \times 2.5°$ grid, for which both day and night measurements were used. Based on cloud information from the EUMETSAT operational processing, the cloud-contaminated scenes are filtered out, i.e. all scenes with a fractional cloud cover higher than 25% are not taken into account. It should be noted that there was an abnormally small

amount of IASI L2 data distributed by EUMETSAT between the 14th of September and the 2nd of December 2010 (Van Damme et al., 2017), and that these data have been removed from the figures and analyses in the following of the paper. For the present study, the data are divided into several time series according to equivalent latitudes (sometimes here referred to as "eqlat") which allow to consider dynamically consistent regions of the atmosphere throughout the globe and to better preserve the sharp gradients across the edge of the polar vortex. The potential vorticity data are daily fields obtained from ECMWF ERA Interim reanalyses, taken at the potential temperature of 530 K. Following the analysis of the potential vorticity contours, we consider 5 equivalent latitude bands in each hemisphere (30°-40°, 40°-55°, 55°-65°, 65°-70°, 70°-90°), plus the intertropical band (30°N - 30°S), with the corresponding potential vorticity contours, in units of $\times 10^{-6}$ K.m$^2$.kg$^{-1}$.s$^{-1}$, being at 2.5 (30°), 3 (40°), 5 (55°), 8 (65°) and 10 (70°) (Figure 1).

## 3 HNO$_3$ time series

The HNO$_3$ time series for the mid to high latitudes are displayed in Figure 2 for the years 2008-2016. Total columns are represented for both north (green) and south (blue curves) hemispheres, for equivalent latitudes bands 40-55°, 55-65°, 65-70°and 70-90°. Also highlighted by shaded areas are the periods during which the southern and northern polar temperatures, taken at 50 hPa (light blue and light green for the 70-90 eqlat band, S and N respectively, and purple for the 65-70 S eqlat band), were equal to or below the polar stratospheric clouds formation threshold (195 K, based on ECMWF temperatures). It should be noted that while this temperature is a widely accepted approximation for the formation threshold for NAT (type I), its actual value can be different depending on the local conditions (Lowe and MacKenzie, 2008; Drdla and Müller, 2010; Hoyle et al., 2013). Also, other forms of PSCs, particularly the type II PSCs (ice clouds) form at a lower temperature of 188 K, corresponding to the frostpoint of water, or 2-3 K below that (e.g. Toon et al. (1989); Peter (1997); Tabazadeh et al. (1997); Finlayson-Pitts and Pitts (2000)).

As a general rule, we find larger concentrations in the northern hemisphere, for the entire latitude range shown here (40-90 eqlat). The hemispheric difference in HNO$_3$ maximum concentrations can be partly attributed to the hemispheric asymmetry of the Brewer-Dobson circulation associated with the many topographical features in the northern hemisphere compared to the southern hemisphere. As a result, the northern hemisphere has a more intense planetary wave activity, which strengthens the deep branch of the Brewer-Dobson circulation. This has also a direct effect on the latitudinal mixing processes, which usually extend into the Arctic polar region, but less into the Antarctic due to a stronger polar vortex (Mohanakumar, 2008; Butchart, 2014).

Beyond the hemispheric asymmetry, we also find that the HNO$_3$ columns are generally larger at higher latitudes, with total column maxima between $3.0\times 10^{16}$ and $3.7\times 10^{16}$ molec.cm$^{-2}$ in the equivalent latitudes bands 70-90°, 65-70° and 55-65°, and lower at around $2.2\times 10^{16}$ molec.cm$^{-2}$ in the 40-55° band, especially for the southern hemisphere. This latitudinal gradient of HNO$_3$ has been previously documented (e.g. Santee et al. (2004); Urban et al. (2009); Wespes et al. (2009); Ronsmans et al. (2016)) and can be explained mainly by the larger amounts of NO$_y$ at high latitudes due to a larger age of air and by the NO$_y$ partitioning favoring HNO$_3$. Another interesting feature observable in Figure 2 is the different behaviour of the three

highest latitude regions with regard to the polar stratospheric clouds formation threshold in the southern hemisphere. The denitrification process that occurs with the condensation and sedimentation of PSCs (see e.g. Wespes et al. (2009); Manney et al. (2011) and Ronsmans et al. (2016) for further details) is obvious in the 70-90 S region (top panel, blue curve in Figure 2), with a systematic and strong decrease in $HNO_3$ total columns (from $3.3\times10^{16}$ to $1.5\times10^{16}$ molec.cm$^{-2}$) starting within 12 to

25 days after the stratospheric temperature reaches the threshold of 195 K (start of the blue shaded areas). The loss of $HNO_3$ thus usually starts around the beginning of June in the Antarctic and the concentrations reach their minimum value within one month. They stay low at $1.4\times10^{16}$ molec.cm$^{-2}$ until mid-November (with quite often a slight gradual increase to $1.7\times10^{16}$ molec.cm$^{-2}$ during the two following months), and start to increase again during January, i.e. between 2.5 and 3 months after the polar stratospheric temperatures are back above the NAT formation threshold. The same pattern can be observed in the

65-70 S equivalent latitude region, with however, a delay of approximately 1 month for the start of the steepest decrease in $HNO_3$ columns, which appears to be more gradual than in the 70-90 S regions (3 months to reach the minimum values, starting in July). The minimum and plateau column values are thus reached by the end of September; they remain higher than at the highest latitudes, with values staying at around $1.7\times10^{16}$ molec.cm$^{-2}$. The delayed and less severe loss of $HNO_3$ in the 65-70 S band confirms that the denitrification process spreads from the center of the polar vortex, where the lowest temperatures are

reached first (McDonald et al., 2000; Santee et al., 2004; Lambert et al., 2016). This spreading from the center also leads to slightly higher concentrations for the maxima in the 65-70 S eqlat band (mean of maxima of $3.26\times10^{16}$, versus $3.11\times10^{16}$ molec.cm$^{-2}$ in the 70-90 S eqlat band). The delayed decrease in $HNO_3$ in the outer parts of the vortex (i.e. in the 65-70 S eqlat band) can thus be attributed to the later appearance of PSCs in this region (see Figure 2 purple shaded areas in second panel). By the end of December, i.e. when the vortex has started breaking down (e.g. Schoeberl and Hartmann (1991); Manney et al.

(1999); Mohanakumar (2008)), the total columns in both eqlat bands homogenize and reach the same range of values ($1.7\times10^{16}$ molec.cm$^{-2}$). If the decrease is slower at 65-70 eqlat, this is not the case for the recovery, with the build-up of concentrations that starts roughly at the same time as for the 70-90 S eqlat band, hence resulting in a shorter period of denitrified atmosphere in the 65-70 S band. These results agree well with previous studies by McDonald et al. (2000) and Santee et al. (2004) for earlier years. However, the recovery of the $HNO_3$ total columns is very slow compared to other species, namely $O_3$, for which

concentrations return rapidly to usual values almost as soon as PSCs disappear. In fact, the $HNO_3$ columns stay low well after the September equinox and are only subject to a slow increase 2 months later (in early December). While more persistent local temperature minima staying below 195 K could explain part of this late recovery, we make the hypothesis that it is due mainly to a combination of two factors: (1) a significant sedimentation of PSCs towards the lower atmosphere during the winter, yielding only small amounts of it to release $HNO_3$ under warmer temperatures (Lowe and MacKenzie, 2008; Kirner

et al., 2011; Khosrawi et al., 2016), and (2) the effective photolysis of $HNO_3$ and $NO_3$ in spring and summer under prolonged sunlight conditions, i.e. mainly at the highest latitudes, which respectively increase the $HNO_3$ sink and reduce the chemical source (because $NO_3$ cannot react with $NO_2$ to produce $N_2O_5$, Solomon (1999); Jacob (2000); McDonald et al. (2000)). The increase observed in March, at the start of the winter, is in turn explainable by a much reduced number of hours of sunlight, implying less photodissociation.

It is worth noting that the two regions previously mentioned (inner and outer vortex) have been observed to behave differently;

the inner vortex (70-90 S) undergoes strong internal mixing whereas the outer vortex (65-70 S), isolated from the vortex core, experiences little mixing of air. This, combined to a cooling of the stratosphere, could lead to increased PSC formation and further ozone depletion (Lee et al., 2001; Roscoe et al., 2012).

Regarding the $HNO_3$ columns in the 55-65 S eqlat band, which comprises the vortex rim, or "collar" (Toon et al., 1989), it is evident from Figure 2, third panel, that it is not affected by denitrification, in agreement with previous observations (e.g. Santee et al. (1999); Wespes et al. (2009); Ronsmans et al. (2016)). In fact, we show that the columns in that band keep increasing when the temperatures at higher latitudes start decreasing, to reach maximum values of about $3.4\times10^{16}$ molec.cm$^{-2}$ in June-July; this is due to a change in the $NO_y$ partitioning towards $HNO_3$, itself due to less sunlight compared to the summer. Also inducing increased concentrations during the winter at high latitudes is the diabatic descent occurring inside the vortex when the temperatures decrease. This downward motion of air enriches the lower stratosphere in $HNO_3$ coming from higher altitude (Manney et al., 1994; Santee et al., 1999), yielding higher column values which are, in this eqlat band, not affected by denitrification. The slow decrease starting in August and leading to minimum values in January is related to the combined effect of increased photodissociation and mixing with the denitrified polar air masses which are no longer confined to the polar regions. Finally, as previously mentioned, the 40-55 S eqlat band records lower column values throughout the year (generally below $2\times10^{16}$ molec.cm$^{-2}$) and a much less pronounced seasonal cycle.

The northern hemisphere high latitudes usually do not experience denitrification, mostly because the temperatures, while sometimes showing local minima below 195 K, rarely reach the PSC formation threshold on broad areas and for long time spans (see Figure 2 for average temperatures, light green vertical areas). A few years stand out, however, with exceptionally low stratospheric temperatures. This is especially the case of the 2011 (Manney et al., 2011), 2016 and, to some extent, 2014 Arctic winters. During these three winters, temperatures reached below the 195 K threshold over a broader area and stayed low during a longer period than usual. Lower concentrations of $HNO_3$ have been recorded in consequence, especially in the northernmost equivalent latitude band (see Figure 2). The winter 2016 recorded in particular exceptionally low temperatures and led to large denitrification and significant ozone depletion (Manney and Lawrence, 2016; Matthias et al., 2016). The denitrification that occurred in the northern polar regions affected a smaller area than what generally happens in the southern hemisphere; in particular the columns in the 65-70 N eqlat band do not show a significant decrease.

Figure 3, which consists in the time series of the zonally averaged distribution of the $HNO_3$ retrieved total columns, illustrates all these features particularly well: it highlights the low and constant columns between -40 and 40 degrees of latitude, the marked annual cycle at mid to high latitudes and the systematic and the occasional (2011, 2014, 2016) loss of $HNO_3$ during the denitrification periods in the high latitudes of the Southern and Northern hemispheres respectively, which are highlighted by the iso-contours of potential vorticity at $\pm\ 10\times10^{-6}$ K.m$^2$.kg$^{-1}$.s$^{-1}$ (dark blue).

In order to give further insights into the interannual variability in polar regions, Figure 4 shows the seasonal cycle for each individual year from 2008 to 2016 for eqlat 70-90 in the northern (top) and the southern (bottom) hemispheres. July and August of 2010 stand out in the Antarctic, with high and variable columns recorded by IASI. This is a consequence of a mid-winter (mid-July) minor sudden stratospheric warming (SSW) event that induced a downward motion of air masses and modified the chemical composition of the atmosphere between 10 and 50 hPa and until at least September (de Laat and van Weele, 2011;

Klekociuk et al., 2011). The principal effect of this sudden stratospheric warming was to reduce the formation of PSCs (which stayed well below the 1979-2012 average (WMO, 2014)) and hence reduce denitrification. This is shown by an initial drop in June, as is usually observed in other years but then by an increase in $HNO_3$ columns when the SSW occurs. These results confirm those previously obtained by the Aura MLS during that winter and reported in the World Meteorological Organization (WMO) Ozone Assessment of 2014 (see Figure 6-3, WMO (2014)). Apart from these peculiarities for the year 2010, all years seem to coincide quite well in terms of seasonality in the southern hemisphere (bottom panel). The timing of the $HNO_3$ steep decrease in particular is consistent from one year to another.

The northern hemisphere high latitudes (top panel) show more interannual variability than in the south, especially during the winter because of the unusual denitrification periods observed in 2011 (purple), 2014 (blue) and 2016 (black) in January (concentrations as low as $2.2 \times 10^{16}$ molec.cm$^{-2}$ in 2016). Contrary to the winter, the summer columns are more uniform from one year to another with values around $2.1 \times 10^{16}$ to $2.8 \times 10^{16}$ molec.cm$^{-2}$.

## 4 Fitting the observations with a regression model

### 4.1 Multi-variable linear regression

In order to identify the processes responsible for the $HNO_3$ variability observed in the IASI measurements, we use a multivariate linear regression model featuring various dynamical and chemical processes known to affect $HNO_3$ distributions. We follow strictly the methodology set-up by Wespes et al. (2016) for similarly investigating the $O_3$ variability. In particular, as in Wespes et al. (2016), we use daily median $HNO_3$ total columns. These are fitted with the following model:

$$HNO_3(t) = cst + y_1.trend + [a_1.cos(\omega t) + b_1.sin(\omega t)] + \sum_{i=2}^{m}[y_i.Y_{\text{Norm,i}}(t)] + \epsilon(t) \tag{1}$$

where $t$ is the day in the time series, $cst$ is a constant term, the $y$ terms are the regression coefficients for each variable, $\omega = 2\pi/365.25$, and $Y_{\text{Norm,i}}(t)$ refer to the chosen explanatory variables $Y$, which are normalized over the period of IASI observations (2008-2016) following:

$$Y_{\text{Norm,i}}(t) = 2(Y(t) - Y_{\text{median}})/(Y_{\text{max}} - Y_{\text{min}}) \tag{2}$$

with $Y_{\text{max}}$ and $Y_{\text{min}}$ the maximum and minimum values of the variable time series (before subtraction of the median, $Y_{\text{median}}$). The terms $a_1$ and $b_1$ in Eq.(1) are the coefficients accounting for the annual variability in the atmosphere. They account mainly for the seasonality of the solar insolation and of the meridional Brewer-Dobson circulation, which is a slow stratospheric circulation redistributing the tropical air masses to extra-tropical regions (Mohanakumar, 2008; Butchart, 2014; Konopka et al., 2015).

The regression coefficients are estimated by the least squares method. The standard error ($\sigma_e$) of each proxy is calculated based on the regression coefficients and is corrected in order to take the autocorrelation uncertainty into account (Knibbe et al., 2014;

Wespes et al., 2016):

$$\sigma_e{}^2 = (\mathbf{Y}^T\mathbf{Y})^{-1} \cdot \frac{\sum[HNO_3 - \mathbf{Y}y]^2}{n-m} \cdot \frac{1+\varphi}{1-\varphi} \tag{3}$$

where $\mathbf{Y}$ is the matrix of explanatory variables of size $n \times m$, $n$ is the number of daily measurements and $m$ the number of fitted parameters. $HNO_3$ is the nitric acid column, $y$ the vector of regression coefficients and $\varphi$ is lag-1 autocorrelation of the residuals.

## 4.2 Iterative selection of explanatory variables

The choice of variables included in the model is made using an iterative elimination procedure; all variables are tested based on their importance for the regression (Mäder et al., 2010). At each iteration, the variable with the largest p-value (and outside the confidence interval of 95%) is removed, until there remain only the variables relevant for the regression, i.e. the ones with a p-value smaller than 0.05. This selection algorithm is applied on each band of equivalent latitude (or grid cell, for the global distributions shown below) and thus yields a different combination of variables, depending on the equivalent latitude region considered.

## 4.3 Variables used for the regression

Given the strong relationship between the $O_3$ and the $HNO_3$ chemistry and variability (Solomon, 1999; Neuman et al., 2001; Santee et al., 2005; Popp et al., 2009) and the novelty in applying such a regression study in an $HNO_3$ dataset, we consider here the major and well known drivers of the total $O_3$ variability, namely: a linear trend, harmonic terms for the annual variability and geophysical proxies for the solar cycle, the QBO, the ENSO phenomenon and for the Arctic (AO) and Antarctic Oscillations (AAO) for the northern and southern hemispheres, respectively. Considering the short length of the time series, however, the linear trend did not yield any significant result and, recalling that the aim of the paper is not to derive long term trends, this aspect will not be discussed further. In addition, a proxy for the volume of polar stratospheric clouds is included to account for the effect of the strong denitrification process during the polar night (cf. Section 3). All the proxies are shown in Figure 5 and described with more details hereafter. The source for each proxy is also provided in Table 1.

### 4.3.1 Solar flux (SF)

As a proxy for the solar activity, we use the 10.7 cm solar flux ($F_{10.7}$). It is a radio flux that varies daily, and correlates to the number of sunspots on the solar disk (Covington, 1948; Tapping and DeTracey, 1990; Tapping, 2013). The data set used here is the adjusted flux that takes the changing earth-sun distance into account. The solar cycle influences directly the partitioning between $NO_y$ (produced by the $N_2O+O^1D$ reaction) and $HNO_3$ through the quantity of sunlight available, and has been known to affect the dynamics and to influence the $O_3$ response in the lower stratosphere (e.g. Hood (1997); Kodera and Kuroda (2002); Hood and Soukharev (2003); Austin et al. (2007)).

### 4.3.2 Quasi-Biennial Oscillation (QBO)

The QBO is one of the main process regulating the dynamics of the tropical atmosphere (e.g. Baldwin et al. (2001); Sioris et al. (2014)). It is driven by vertically propagating gravity waves, which lead to an oscillation between stratospheric winds blowing from east (easterlies) and from west (westerlies), with a period of about 28-29 months (e.g. Hauchecorne et al. (2010); Schirber (2015)). Its effect on the distribution of chemical species is significant, especially in equatorial regions where both a direct effect due to the changing winds and an indirect effect via its influence on the Brewer-Dobson circulation, affect, for example, the distribution of ozone (e.g. Lee and Smith (2003); Mohanakumar (2008); Frossard et al. (2013); Knibbe et al. (2014)). Two monthly time series of QBO at two different pressure levels (30 hPa and 10 hPa) from ground-based measurements in Singapore have been considered for the present study, in order to take into account the differences in phase and shape of the QBO signal in the upper and lower stratosphere.

### 4.3.3 Multivariate ENSO Index (MEI)

The Multivariate ENSO Index is a metric that quantifies the strength of the El Niño-Southern Oscillation; it is computed based on the measurement of six variables over the tropical Pacific: sea-level pressure, zonal and meridional winds, sea surface temperature, surface air temperature and cloudiness fraction (Wolter and Timlin, 1993, 1998). The ENSO phenomenon, even though it is a tropospheric process (mainly sea surface temperature contrasts), also affects stratospheric circulation. Previous studies have shown the impact of El Niño/La Niña oscillation on the stratospheric transport processes and the generation of Rossby waves, in turn modulating the strength of the polar vortex (e.g. Trenberth et al. (1998); Newman et al. (2001); Garfinkel et al. (2015)) and affecting $O_3$ in the stratosphere (e.g. Randel et al. (2009); Lee et al. (2010); Randel and Thompson (2011)).

### 4.3.4 Arctic Oscillation and Antarctic Oscillation

The AO and AAO are included in the regression in order to represent the atmospheric variability observed in the northern and southern hemispheres, respectively (Gong and Wang, 1999; Kodera and Kuroda, 2000; Thompson and Wallace, 2000). They are constructed from the daily geopotential height anomalies in the 20-90° region, at 1000 mb (for the northern hemisphere) and 700 mb (for the southern hemisphere). Each index (AO or AAO) is considered only in the hemisphere it is related to, while both indices are included for equatorial latitudes. The impact of these oscillations on $O_3$ distributions has been demonstrated in several studies (e.g. Rieder et al. (2013); Wespes et al. (2016)). We may expect similar influence on the $HNO_3$ distributions, particularly because, even though they are tropospheric features, their phase and intensity affect the atmospheric circulation, and in particular the Brewer-Dobson Circulation, up to the stratosphere (Miller et al., 2006; Chehade et al., 2014).

### 4.3.5 Volume of Polar Stratospheric Clouds (VPSC)

The very low temperatures recorded during the winter in the polar stratosphere inside the vortex lead to the formation of PSCs, which are composed of nitric acid di- or trihydrates (NAD or NAT), supercooled ternary $HNO_3/H_2SO_4/H_2O$ solutions (STS) or water ice ($H_2O$) (e.g Wang and Michelangeli (2006); Drdla and Müller (2010)). Here, we consider for the PSCs only the

NAT particles (HNO$_3$.(H$_2$O)$_3$), which are ubiquitous (and often mixed with STS) (Voigt et al., 2000; Pitts et al., 2009; Lambert et al., 2016). The other forms of PSCs are expected to influence the variability in gas-phase HNO$_3$ to a much lesser extent (von König et al., 2002).

The proxy we use here for the NAT is the volume of air below $T_{NAT}$ (195 K), which depends on nitric acid concentrations,

water vapor and pressure (Hanson and Mauersberger, 1988; Wohltmann et al., 2007). The temperatures needed to compute that quantity are based on ERA-Interim reanalyses and the HNO$_3$ and H$_2$O profiles are taken north and south of 70° from an MLS climatology. The proxy is calculated with a supersaturation of HNO$_3$ over NAT of 10, roughly corresponding to 3K supercooling (Hoyle et al., 2013; Lambert et al., 2016; Wohltmann et al., 2017). It should be noted that this proxy was not included in the regression outside of the polar regions. Inside the polar regions (eqlat bands 70-90 north and south), it was

included and subject to the selection algorithm.

Finally, it is worth to note that, for the sake of completeness, proxies accounting for the potential vorticity (PV) and for the Eliassen-Palm flux (EPflux) were also tested in order to take into account more precise patterns of the stratospheric dynamics and the Brewer-Dobson circulation. Also, various levels for the QBO were tested. However, none of these proxies lead to a significant improvement of the residuals or the correlation coefficients, and their signal is therefore embedded here in the

harmonic terms. For these reasons, they will not be discussed further.

## 4.4    Results

The results are presented in two ways: first, latitudinally averaged time series (eqlat bands) are used to analyze the performances of the fit in terms of correlation coefficients and residuals, with a focus on polar regions. The performance of the model is then analysed in terms of global distributions (with the regression applied to every $2.5° \times 2.5°$ grid cell) and the spatial distribution

of the fitted proxies is detailed.

### 4.4.1    HNO$_3$ fits for equivalent latitudes bands

For each eqlat band, the variables retained by the selection procedure (see Section 4.2) are listed in Table 2. Most variables are retained everywhere, except for the solar flux which is rejected in the polar latitudes (70-90 N and S). The QBO30 is also excluded in the southern polar regions (65-90 S) and the MEI in the northern polar regions (65-90 N). Finally, the AO and

AAO are excluded in the 65-70 N and in the 70-90 S bands, respectively.

The results from the multivariate regression are presented in Figure 6 for each band of equivalent latitude. The model reproduces well the measurements, with correlation coefficients between 0.81 (in the 30-40 N eqlat band) and 0.94 (in the 70-90 S eqlat bands). Most major features (seasonal and interannual variabilities) are reproduced by the regression model. The residuals range between $1.74 \times 10^{10}$ and $9.44 \times 10^{15}$ molec.cm$^{-2}$, with better results for the 30N-30S equivalent latitude band (Root Mean

Square Error (RMSE) of $2.39 \times 10^{14}$ molec.cm$^{-2}$) and worse fits for the 65-70 S band (RMSE of $2.41 \times 10^{15}$ molec.cm$^{-2}$). Following the comparison between the fits and the observational data, some features can be highlighted:

– The high daily variability recorded in the data during the winter for both polar regions is not captured very well by the regression fit. Indeed, we find that the residuals are largest in this period, especially in the southern hemisphere during the denitrification period of each year (from June until September approximately), mostly because of the high variability of the vortex itself. There, we find an average standard deviation of $1.44\times10^{16}$ molec.cm$^{-2}$ (average of the standard deviation during the denitrification periods over the 8 years of observation), as opposed to a mean standard deviation of $8.30\times10^{15}$ molec.cm$^{-2}$ for the periods between the denitrification seasons. In the northern hemisphere, the day-to-day variability is largest during winter as well, when the vortex builds up, and this causes larger residuals for the corresponding months (see December through March of each year, top left panel of Figure 6, with an average standard deviation of $7.97\times10^{15}$ molec.cm$^{-2}$ to be compared to $7.26\times10^{15}$ molec.cm$^{-2}$ for the other months). It is important to stress that these larger residuals are obtained in the polar regions despite the fact that a VPSC proxy was used. In Figure 7 we show, however, that the regression model would perform worse in polar regions if that proxy is neglected, as also discussed below.

– Even though the high variability during the denitrification periods is not reproduced exactly, the amplitude of the decrease in HNO$_3$ occurring in the Southern polar region is captured accurately by the regression model. Figure 7 shows a zoom of Figure 6 to better highlight the model performance during the denitrification periods; the regression was tested without (top panels) and with (middle panels) the VPSC proxy, for the 70-90 N (left panels) and the 70-90 S (right panels) eqlat bands. The steep slope observed at the start of the low temperatures is captured by the model when the proxy for the VPSC is included (Figure 7) and the correlation coefficients are improved for both hemispheres (from 0.83 to 0.86 in the 70-90 N and from 0.84 to 0.94 in the 70-90 S eqlat band). In the 65-70 S eqlat band however, as previously described in Section 3, the HNO$_3$ columns continue to increase after the formation of PSCs has started in the 70-90 S eqlat band. This translates to a lag between the observations in the 65-70 S eqlat band and the fit in which the drop of HNO$_3$ concentrations happens earlier than in the IASI observations. This is explained by the fact that the VPSC proxy is based on temperatures and composition poleward of $70°$. It induces a lower correlation coefficient (0.87) and higher RMSE ($2.41\times10^{15}$ molec.cm$^{-2}$). A proxy adapted to this eqlat band should be used in further studies in order to represent the conditions in that particular region of the vortex.

– The high maxima seen in the IASI time series, mostly from mid-April through the end of May in the Southern hemisphere, and from mid-December through early February in the Northern hemisphere, are not that well reproduced by the regression model. In fact, the model fails to capture the highest columns during the winters of each hemisphere. In the same way, a few pronounced lows recorded by IASI, especially those in the Northern polar regions (mid-June to early October 2014 and 2016, for instance) are not captured by the model.

Figure 8 shows the regression coefficients of each variable in each equivalent latitude band (top panel). The two bottom panels show the signal of the fitted proxies, calculated by multiplying the proxy by its regression coefficient. Only the variables retained by the selection algorithm are shown and discussed. From the top panel of Figure 8, it can be seen that all proxies are

significant, with errors smaller than the coefficients for all eqlat bands. It is clear that the annual variability is predominant at all latitudes. From the two bottom panels, we also see the large influence of the VPSC in the regression for the polar regions. Their signal is, as expected, larger in the southern hemisphere where it reaches $-1.3 \times 10^{16}$ molec.cm$^{-2}$, which is to be compared to maximum values around $-0.4 \times 10^{16}$ molec.cm$^{-2}$ in the northern hemisphere. A noteworthy difference is found for the year

2016 where the VPSC signal reached $-0.7 \times 10^{16}$ molec.cm$^{-2}$ during the exceptionally cold Arctic winter. While the PSCs have significantly affected the $HNO_3$ distributions in the winters 2011, 2014 and 2016 in the Arctic, their influence during other years may contribute to the high variability recorded in the observations (see first highlighted feature above). Other proxies show relatively large signals and their global distribution will be discussed further in Section 4.4.3.

### 4.4.2   Global model assessment with regard to the $HNO_3$ variability

To assess the model ability to reproduce the measurements, the top panel of Figure 9 shows the percentage of the $HNO_3$ variability seen by IASI that is explained by the regression model. The fraction is calculated as the difference between the standard deviation of the fit and of the observations [$\sigma$ ($HNO_3^{\text{fit}}$(t))/ $\sigma$ ($HNO_3^{\text{IASI}}$(t)) $\times$ 100] and is expressed as a percentage. We find that much of the observed variability can be explained by the model in the southern hemisphere (generally between 50 and 80 %). The southern mid-latitudes and the polar regions are in particular well modeled (70-80 %), except in Antarctica

above the ice shelves. The northern hemisphere $HNO_3$ variability is reasonably well explained by the model, particularly above 40° of latitude, with percentages ranging between 50 and 80 %, although some continental areas (Northern part of inner Eurasia above Kazakhstan and the west Siberian plains) stand out with percentages below 40 %. The region with the largest unexplained fraction of variability is the intertropical band extending as far as 40° north. There, the fraction of $HNO_3$ variability explained by the model reaches values as low as 20 %. These regions of low explained variability coincide quite well with the

regions where a high lightning activity is found, which produces large amount of $NO_x$ in the troposphere (Labrador et al., 2004; Sauvage et al., 2007; Cooper et al., 2014). While the IASI instrument is usually not sensitive to tropospheric $HNO_3$, it was found that large amounts of tropospheric $HNO_3$ in the tropics could be detected, mainly because of the lower contribution of the stratosphere in that region, and because the $NO_x$ produced by lightning are released in the high troposphere, where IASI has still reasonable sensitivity. This could thus explain why the model is missing some of the variability recorded in the

observational data. Another cause for the discrepancies between the observations and the model could be unaccounted sinks of $HNO_3$, such as deposition in the liquid or solid phase and scavenging by rain. It should be noted that a small area of high explained variability is observed in Africa, just south of the equator. The variability in this region is unexpectedly high in the IASI time series (Figure 10) and we suggest that it could be influenced by biomass burning emissions of $NO_2$, and subsequent oxidation to $HNO_3$ with a delay of about 2 months (Figure 10) (Scholes et al., 1996; Barbosa et al., 1999; Schreier et al.,

2014). Indeed, the large vegetation fires of Africa every year around July emit the largest amounts of $NO_x$ (compared to large fires of South America, Australia and southeast Asia). Their influence translates to an overrepresentation of the annual term (up to $-2 \times 10^{15}$ molec.cm$^{-2}$) in the fitted model (although not clearly visible in Figure 11 because of the color scale chosen). This larger contribution of the annual variability thus yields a better agreement between the observations and the model in the tropical band, however missing some of the interannual variability due to these fires.

The bottom panel of Figure 9 depicts the global distribution of the RMSE of the regression expressed as a percentage. The errors are small everywhere (between 10 and 20 %) except in the southern hemisphere above Antarctica, and particularly above the ice shelves (mainly the Ross and Ronne ice shelves). We also find higher values above the large desert areas (the Sahara, the Arabian, the Turkestan and the Australian deserts) as well as off the west coasts of south Africa and South America where persistent low clouds occur. These regions of low clouds or characterized by sharp or seasonally varying emissivity features are known to cause problems for the retrieval of $HNO_3$ using the IASI spectra (Hurtmans et al., 2012; Ronsmans et al., 2016).

### 4.4.3 Global patterns of fitted parameters

Figure 11 shows the global distributions of the regression coefficients obtained after the multivariate regression, expressed in molec.cm$^{-2}$. All the variables are shown, with the areas where the proxy was not retained left blank. The contribution of each proxy to the $HNO_3$ variability was also calculated for each grid cell as $[\sigma(X_i)/\sigma(HNO_3{}^{IASI}) \times 100]$ with $X_i$ referring to each of the i explanatory variables $X$, and expressed in %. Note that, although the distributions of the contribution of each proxy are not shown as a Figure, the calculated percentage values are used in the following discussion (next 3 subsections) to quantify the influence of the fitted parameters.

**The annual cycle**

The annual cycle, represented by the terms $a_1$ and $b_1$, shows large regression coefficients (Figure 11) and holds the largest part of the variability globally (up to $70\%$ in the northern and southern mid to high latitudes), as was previously evidenced in Figure 8 (top panel). While the Brewer-Dobson circulation, which is embedded in these harmonic terms, influences to some extent the $HNO_3$ variability (through its influence on the conversion of $N_2O$ to $NO_y$ in the tropics and through the transport of $NO_y$-rich air masses towards the polar regions and subsequent transformation into $HNO_3$), the influence of the seasonality of the solar insolation is also likely to largely influence the annual seasonality, especially in the mid- to high latitudes. The increasing columns recorded during the winter in both polar regions can be explained by the combination of three processes: first, at low temperatures, $HNO_3$ is formed by heterogeneous reactions between $N_2O_5$ and $H_2O_{aerosol}$ and between $ClONO_2$ and $H_2O_{aerosol}$ or $HCl_{aerosol}$, which add to the main source gas-phase reaction $OH+NO_2+M{\rightarrow}HNO_3$. Second, while the source reactions of $HNO_3$ are still active, the loss reactions ($HNO_3$ photolysis and its reaction with OH) are significantly slowed down during the winter (Austin et al., 1986; McDonald et al., 2000; Santee et al., 2004). And third, as is mentioned is Section 3, with the decrease of the temperatures in the polar stratosphere, the winds inside the polar vortex gain intensity and induce a strong diabatic downward motion of air with little latitudinal mixing across the vortex boundary. This descending air from the upper stratosphere enriches the lower stratosphere in $HNO_3$ (Schoeberl and Hartmann, 1991; Manney et al., 1994; Santee et al., 1999).

**The solar cycle, MEI, AO/AAO and QBO**

The solar flux, ENSO index and Arctic and Antarctic Oscillation (Figure 11) all have a similar influence in terms of magnitude (between $-2.5 \times 10^{15}$ and $2.5 \times 10^{15}$ molec.cm$^{-2}$), although with different spatial patterns. The influence of the solar flux is positive in the northern polar latitudes and at the edge of eastern Antarctica, as well as above the Indian Ocean and Australia. It

is close to zero or negative elsewhere. Previous studies showed that ozone changes due to the solar cycle are largest in the low stratosphere (Hood, 1997; Soukharev and Hood, 2006), which corresponds to the altitude of maximum sensitivity for HNO$_3$. Our results for the mid to high latitudes suggest opposite behaviour for HNO$_3$ (as was also reported for O$_3$ by Wespes et al. (2017)). However, the positive contribution of the solar cycle on the HNO$_3$ variation in the tropical stratosphere is in line with the low-latitude O$_3$ response previously reported (Soukharev and Hood, 2006; McCormack et al., 2007; Frossard et al., 2013;

Maycock et al., 2016). Note also that the strong negative signal observed above western Antarctica is most probably due to the drawback of using for all seasons a constant emissivity for ocean surfaces (e.g. even when the ocean becomes frozen). For this reason, the regression coefficients in this area will not be discussed further.

The MEI shows a negative signal above the northern polar regions and in the eastern parts of the Pacific and Atlantic (especially west of South Africa) Oceans. A positive signal is observed above Australia and above the southern polar regions. Overall, the

MEI influence is quite small, which is not surprising considering that it affects mostly the tropospheric circulation, where IASI is less sensitive. Its signature is nonetheless visible and significant in the eastern Pacific, where it contributes to up to 30% of the HNO$_3$ variability, and in the mid-latitudes of the northern hemisphere. The east-west gradient is in good agreement with chemical and dynamical effects of El Niño on O$_3$, and with previous studies that showed the same patterns for the influence of the MEI on O$_3$ (Hood et al., 2010; Rieder et al., 2013; Wespes et al., 2017).

The arctic oscillation (AO) signal is stronger, especially above the Atlantic Ocean, with a positive signal above eastern Canada and Greenland and between the north of eastern Africa and Florida. Except for those two regions, the AO contributes at mid to high latitudes of the northern hemisphere with a negative signal, which contributes for $10 - 20\%$ to the HNO$_3$ variability. The corresponding proxy for the southern hemisphere (AAO) is also significant, with a strong positive signal above the vortex rim and a negative signal above Antarctica. These results are in agreement with previous studies that showed that, for O$_3$, both the

arctic and antarctic oscillations (also called "annular modes") are leading modes of variation in the extratropical atmosphere (Weiss et al., 2001; Frossard et al., 2013; de Laat et al., 2015; Wespes et al., 2017). They strongly influence the circulation up to the lower stratosphere and represent, particularly in the southern hemisphere, the fluctuations in the strength of the polar vortex (Thompson and Wallace, 2000; Jones and Widmann, 2004; van den Broeke and van Lipzig, 2004). This further shows the similarity in the behaviour of O$_3$ and HNO$_3$.

The QBO has a generally small influence on the distributions with, however, some contribution (up to 30%) in the equatorial band as expected (Baldwin et al., 2001; Solomon et al., 2014). As previously mentioned, several tests have been performed (not shown here) with the QBO taken at other pressure levels in the atmosphere (namely 20 and 50 hPa), and similar results have been obtained. Even though the QBO is a tropical phenomenon, its effects extend as far as the polar latitudes, through the modulation of the planetary Rossby waves (e.g. Holton and Tan (1980); Baldwin et al. (2001)). Because there are more

topographical features in the northern hemisphere than in the southern hemisphere, these waves have a larger amplitude and can influence the polar stratospheric temperatures and hence the vortex formation. While the exact mechanism for the extratropical influence of the QBO is not exactly understood (Garfinkel et al., 2012; Solomon et al., 2014), it seems the large positive and negative signals observed in the northern high latitudes in Figure 11 can indeed be attributed to this modulation of the Rossby waves by the oscillation in the meridional circulation. This was also observed for $O_3$ by e.g. Wespes et al. (2017).

**VPSC**

The annual cycle, which is the dominant factor for $HNO_3$ variability at all latitudes, leading to the build-up of concentrations during the winter, is interrupted in the southern polar regions, particularly in the 70-90 S eqlat band (see also Figure 8), by the condensation and subsequent sedimentation of PSCs. The VPSC proxy, reflecting the volume of air below $T_{NAT}$, has a strong anti-correlated effect on $HNO_3$ columns, which decrease (negative values) with increasing VPSC (e.g. Wang and Michelangeli (2006); Lowe and MacKenzie (2008); Kirner et al. (2015)). The signal of the VPSC proxy is thus, as expected, negative everywhere (in the polar regions considered), with values around -6×$10^{15}$ molec.cm$^{-2}$. When looking at their contribution, we find that the PSCs account for a larger part of the $HNO_3$ variability ($40 - 60\%$) in the southern hemisphere, where the influence of denitrification is indeed expected to be more important, compared to the northern hemisphere (maxima of $40\%$), as discussed in Section 3 with the analysis of Figure 2. The small areas with a positive signal appear to be non significant (see grey crosses).

## 5 Conclusions

Time series of $HNO_3$ total columns retrieved from IASI/Metop between 2008 and 2016 have been presented and analyzed in terms of seasonal cycle and global variability. The analysis was conducted in terms of equivalent latitudes (here calculated on the basis of potential vorticity) and focused mainly on high latitude regions. We have shown that the IASI instrument captures the broad patterns of the seasonal cycles at all latitudes but also year-to-year specific behaviours. The systematic denitrification process occurring every winter-spring in the southern hemisphere shows up unambiguously in the time evolutions and the use of equivalent latitudes has enabled to isolate the regions affected based on the dominating stratospheric dynamical regimes. Three distinct zones within the polar regions were in particular separated; 1) the inner polar region (70-90 S), where the denitrification starts the earliest and where the $HNO_3$ columns reach their lowest values for the longest period; 2) the outer part of the polar vortex (65-70 S), where the $HNO_3$ columns drop occurs 1 month later and the minimum concentrations do not reach such low levels; 3) the polar vortex edges (55-65 S), where the columns follow a more normal annual cycle, with maxima around July, forming a collar of high columns around the denitrified vortex. The IASI-derived $HNO_3$ distributions also reflect the denitrification periods in the northern hemisphere, during the exceptionally cold winters of 2011, 2014 and 2016.

The $HNO_3$ time series have been successfully fitted with multivariate regressions in order to identify the various factors responsible for the variability in the observations. To the best of our knowledge, this is the first time that such regression models are applied to the $HNO_3$ time evolution. A specific set of explanatory variables was retained for each equivalent latitude band

following an iterative procedure, according to the influence of each of these variables in the regression. The regression model allowed good representation of the IASI observations in most cases (correlation coefficients between 0.81 and 0.94). However, the variability recorded in the tropics could not be reproduced that well, with only about 20 to 40 % correctly accounted for. The regression for other parts of the globe yielded better results, especially in the southern polar regions, where a high per-
centage (60-80 %) of the observed variability is reproduced by the regression. Generally, it was found that the annual cycle is the factor responsible for the largest part of the variability, showing a hemispheric pattern. The Brewer-Dobson circulation, and also the solar insolation seasonality, which are embedded in the harmonic terms, seem to be the main drivers of variability, with the Brewer-Dobson circulation carrying $NO_y$ towards the poles and both processes bringing the $HNO_3$ concentrations to their maxima during the local winter when production is enhanced and destruction inhibited. We show interestingly that
the polar stratospheric clouds contribute as the second most important driver of the variability of $HNO_3$ in the southern polar latitudes (65-90 S). The influence of PSCs is as expected less marked in the northern hemisphere, but accounting for PSCs still significantly improves the model-to-observation agreement especially during the colder northern winters (R from 0.83 to 0.86). While we feel that the VPSC proxy used here for the PSCs (including only the NAT) is generally good, it is not excluded that adding other forms of PSCs would further improve the model. In any case, the present work shows the potential of using the
IASI measurements to study in depth the polar denitrification processes.

Going towards to the mid- and tropical latitudes, the annual cycle is still prominent, but the relative influence of the QBO increases. Most of the weak seasonality revealed by IASI in the tropical regions is explained by the annual cycle (as well as a potential contribution of African fires and of lightning for additional $NO_x$ sources), the QBO and the MEI.

More generally, this study shows that the IASI data allow a good analysis and understanding of the $HNO_3$ variability in the
atmosphere. The measurements are made with exceptional spatial and temporal sampling, which allows a detailed analysis of the polar regions throughout the entire year. The amount of data allows for a thorough monitoring of the processes regulating the $HNO_3$ variability, such as the denitrification processes in the southern polar regions, or the seasonal variability in the trop-ical regions. The IASI $HNO_3$ time series will soon be extended with the launch of Metop C in September 2018, which should further improve the regression model. As shown here by the still significant residuals at some periods and locations, other
factors could also probably be included to acquire a full and coherent representation of the $HNO_3$ total columns variability.

*Acknowledgements.* IASI has been developed and built under the responsibility of the "Centre National d'Etudes Spatiales" (CNES, France). It is flown on board the Metop satellites as part of the EUMETSAT Polar System. The IASI L1 data are received through the EUMETCast near-real-time data distribution service. The research was funded by the F.R.S.-FNRS, the Belgian State Federal Office for Scientific, Tech-nical and Cultural Affairs (Prodex arrangement 4000111403 IASI.FLOW) and EUMETSAT through the Satellite Application Facility on
Atmospheric Composition Monitoring (ACSAF). The authors would like to thank Ingo Wohltmann for the VPSC proxy and for useful dis-cussions. G. Ronsmans is grateful to the "Fonds pour la Formation à la Recherche dans l'Industrie et dans l'Agriculture" of Belgium for a PhD grant (Boursier FRIA). Cathy Clerbaux is grateful to CNES for financial support.

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

**Table 1.** Proxies used for the regressions and their source

| Proxy | Description | Source |
|---|---|---|
| SF | Solar Flux at 10.7 cm | NOAA National Center for Environmental Information |
| | | https://www.ngdc.noaa.gov/stp/solar/flux.html |
| QBO | Quasi Biennial Oscillation | Free University of Berlin |
| | index at 10 and 30hPa | http://www.geo.fu-berlin.de/en/met/ag/strat/ |
| | | produkte/qbo/index.html |
| MEI | Multivariate ENSO Index | NOAA Earth System Research Laboratory |
| | | http://www.esrl.noaa.gov/psd/data/climateindices/ |
| VPSC | Volume of Nitric Acid Trihydrates | Dr. Ingo Wohltmann at AWI |
| | formed in the stratosphere | (personal communication) |
| AO & AAO | Arctic & Antarctic | NOAA Earth System Research Laboratory |
| | oscillation indices | http://www.esrl.noaa.gov/psd/data/climateindices/ |

**Table 2.** Set of variables retained by the selection algorithm for each equivalent latitude band.

| 70-90S | 65-70S | 55-65S | 40-55S | 30-40S | 30-30 | 30-40N | 40-55N | 55-65N | 65-70N | 70-90N |
|--------|--------|--------|--------|--------|--------|--------|--------|--------|--------|--------|
|        | SF     | SF     | SF     | SF     | SF     | SF     | SF     | SF     | SF     |        |
| QBO10  | QBO10  | QBO10  | QBO10  | QBO10  | QBO10  | QBO10  | QBO10  | QBO10  | QBO10  | QBO10  |
|        |        | QBO30  | QBO30  | QBO30  | QBO30  | QBO30  | QBO30  | QBO30  | QBO30  | QBO30  |
| COS1   | COS1   | COS1   | COS1   | COS1   | COS1   | COS1   | COS1   | COS1   | COS1   | COS1   |
| SIN1   | SIN1   | SIN1   | SIN1   | SIN1   | SIN1   | SIN1   | SIN1   | SIN1   | SIN1   | SIN1   |
| MEI    | MEI    | MEI    | MEI    | MEI    | MEI    | MEI    | MEI    | MEI    |        |        |
| VPSC   | VPSC   |        |        |        |        |        |        |        | VPSC   | VPSC   |
|        | AAO    | AAO    | AAO    | AAO    | AO/AAO | AO     | AO     | AO     |        | AO     |

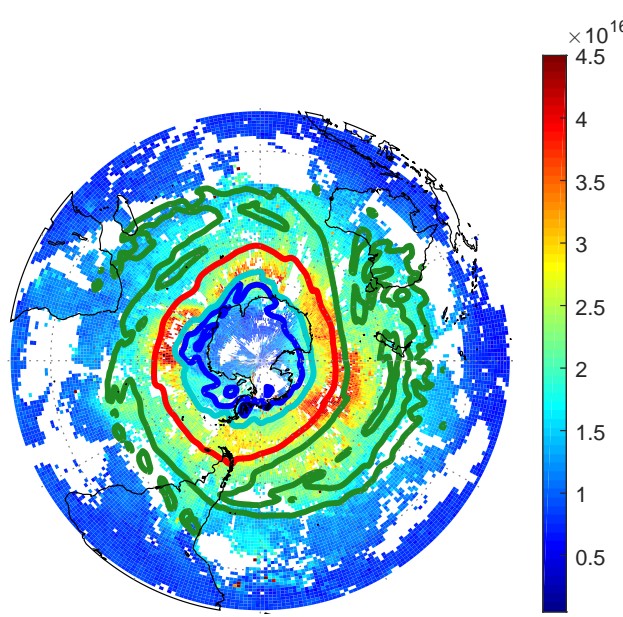

**Figure 1.** Example of equivalent latitude contours for -70 (blue), -65 (light blue), -55 (red) and -40 (green) equivalent latitudes. The background colors are $HNO_3$ total columns (daily mean for 21.07.2011, in molec.cm$^{-2}$).

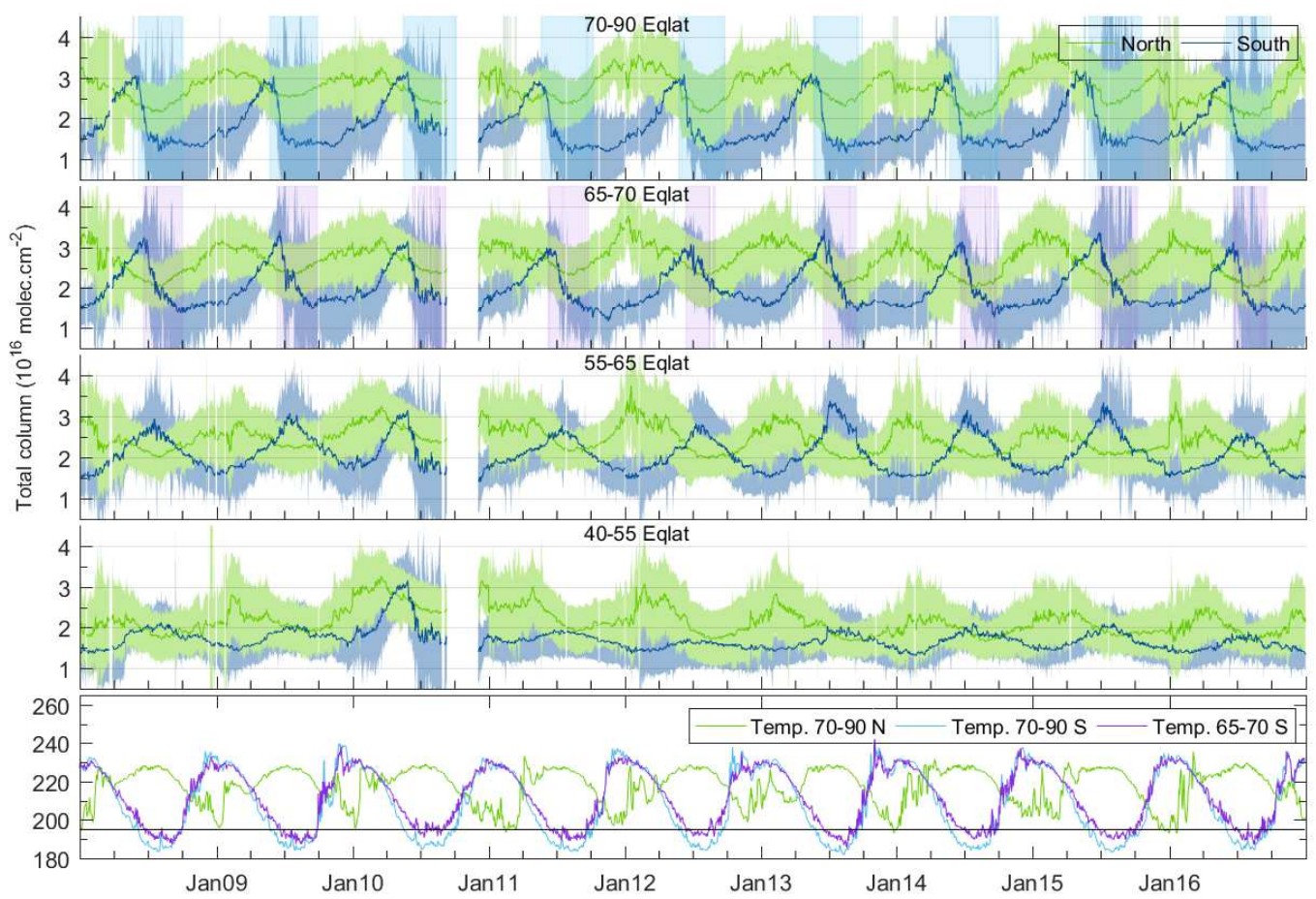

**Figure 2.** Four top panels: HNO₃ total columns time series for the years 2008-2016, for equivalent latitude bands 70-90, 65-70, 55-65 and 40-55, north (green) and south (blue). Vertical shaded areas are the periods during which the average temperatures are below $T_{NAT}$ in the north (green) and south (blue) 70-90° band, and in the south (purple) 65-70° band. Note that the large period without data in 2010 is when there was a low amount of data distributed by EUMETSAT (see Section 2). Bottom panel: Daily average temperatures time series (in K) taken at the altitude of 50 hPa for the equivalent latitude bands 70-90° North (green) and South (blue) and 65-70° South (purple). The horizontal black line represents $T_{NAT}$, i.e. the 195 K line.

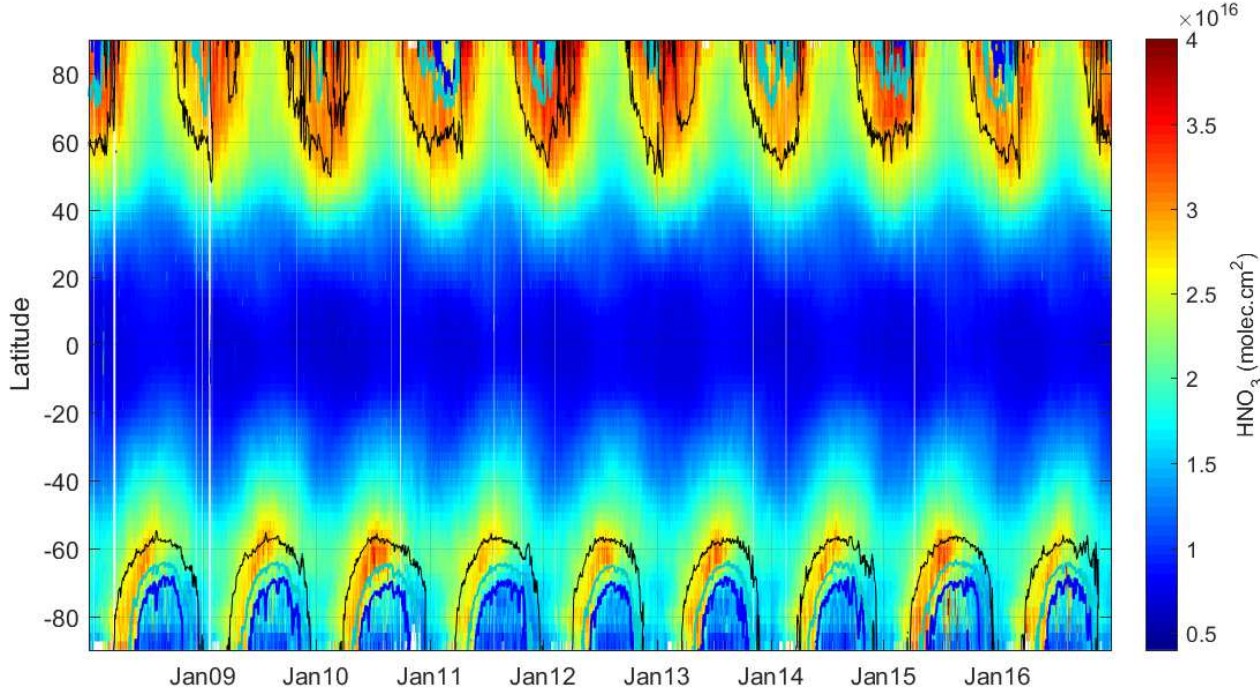

**Figure 3.** Zonally averaged daily HNO₃ total columns distribution over 2008-2016, expressed in molec.cm⁻². The lines represent potential vorticity contours at a potential temperature of 530 K (5 (black), 8 (cyan) and 10 (blue)×10⁻⁶ K.m².kg⁻¹.s⁻¹) which correspond to the equivalent latitudes contours illustrated in Fig 1.

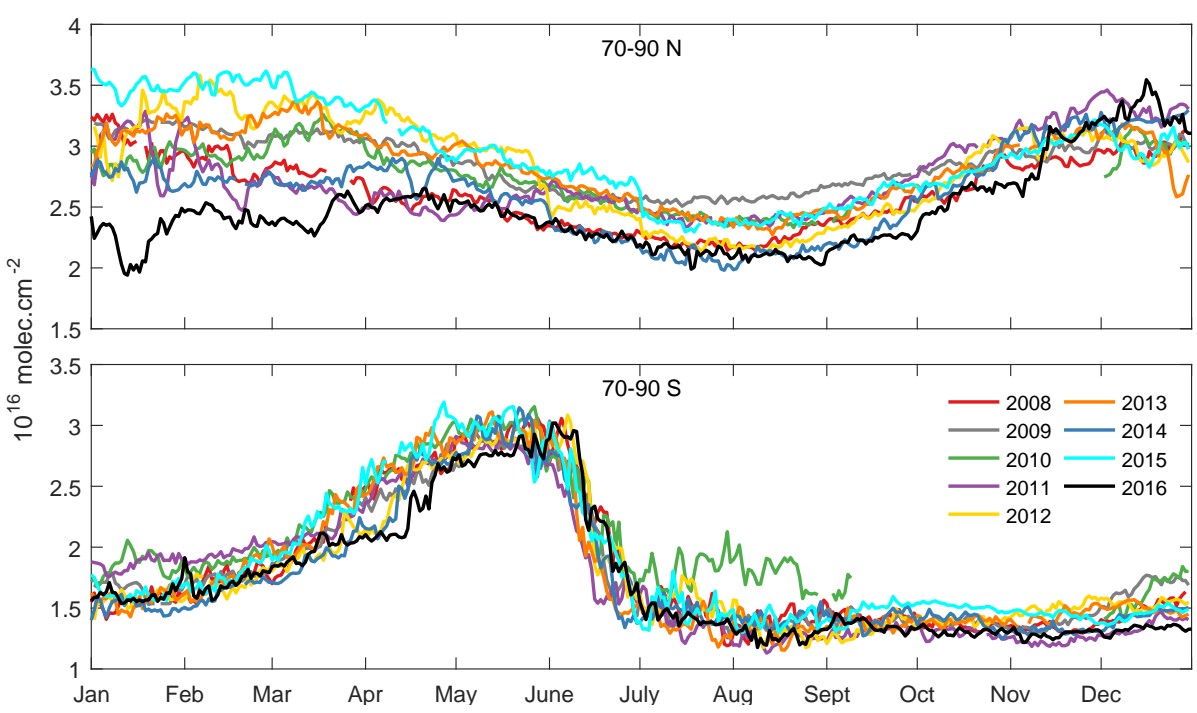

**Figure 4.** For northern (top) and southern (bottom) 70-90 equivalent latitude bands: HNO$_3$ total columns time series for the years 2008 to 2016 in molec.cm$^{-2}$. Note that the y-axis limits differ between the two plots.

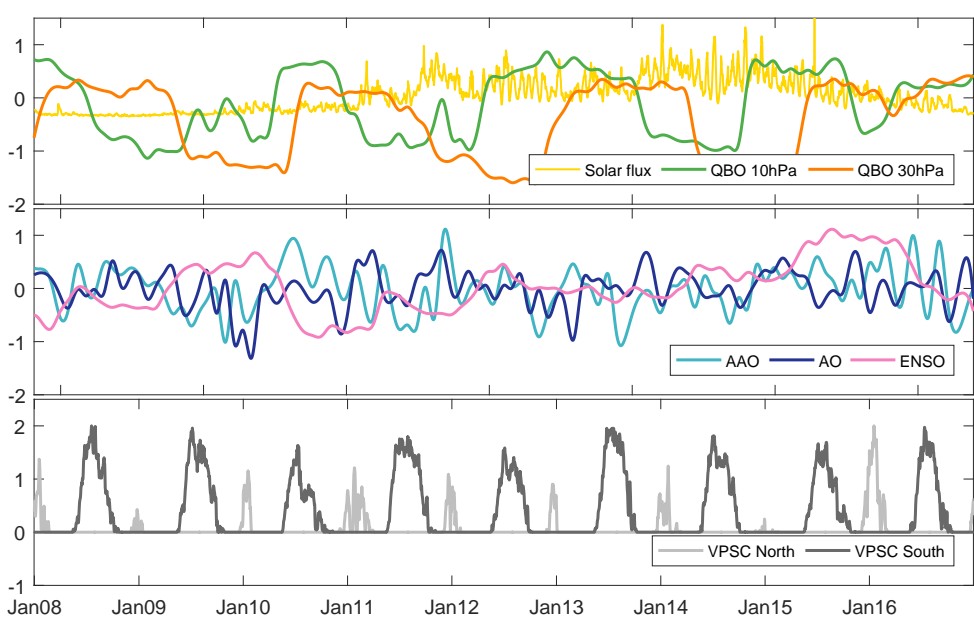

**Figure 5.** Normalized proxies over the IASI observations period (2008-2016). Top panel: Solar flux (yellow), QBO at 10 hPa (green) and QBO at 30 hPa (orange). Middle panel: Antarctic Oscillation (light blue), Arctic oscillation (dark blue) and Multivariate ENSO Index (MEI, pink). Bottom panel: VPSC proxy in the northern hemisphere (light grey) and in the southern hemisphere (dark grey).

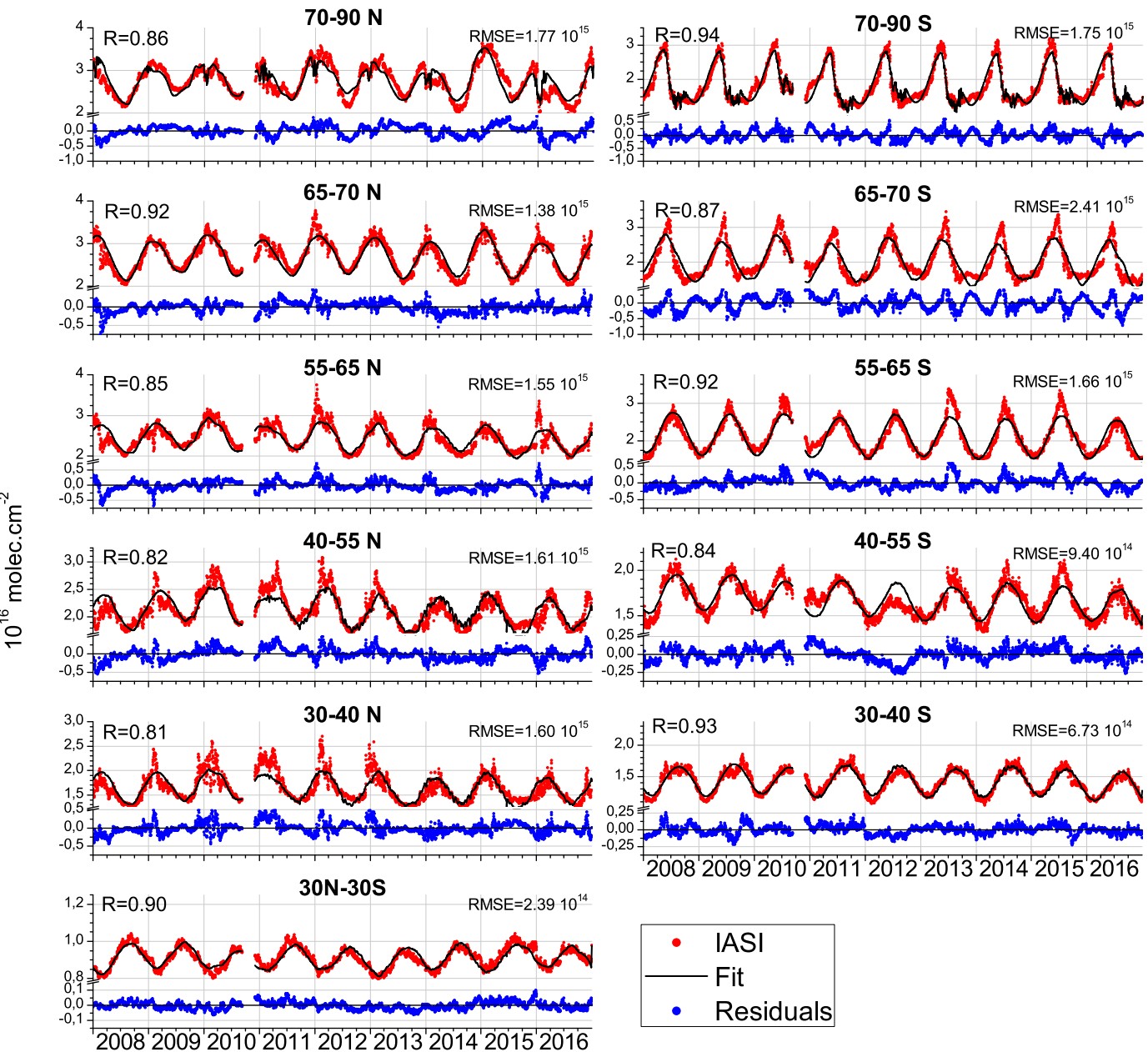

**Figure 6.** IASI $HNO_3$ total columns (red dots) for each equivalent latitude bands and the associated fitted model (black curves). The residuals are in blue, and the horizontal black line represents the zero residual line. For each equivalent latitude band, the correlation coefficient (R) between the observations and the model fit is given in the top left corner, and the root mean square error (RMSE) in the top right corner.

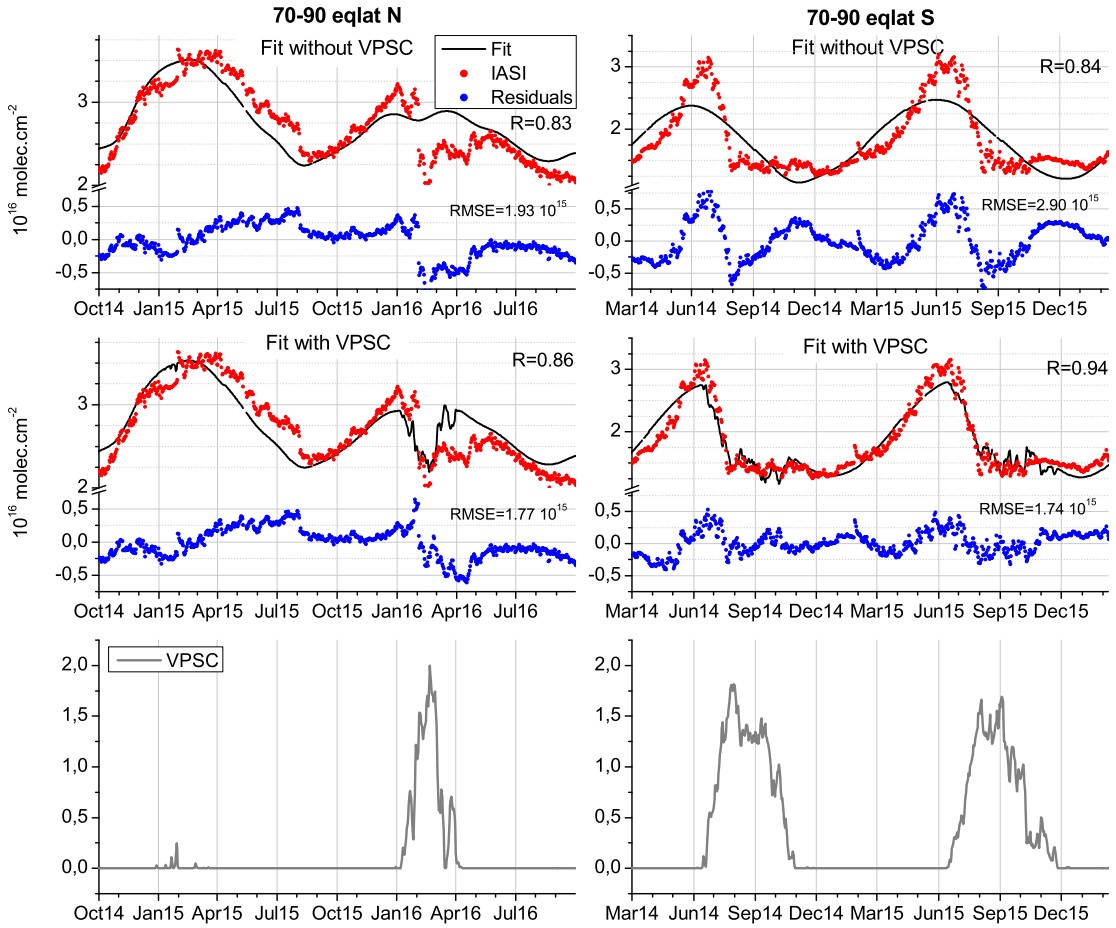

**Figure 7.** For north (left) and south (right) equivalent latitude bands 70-90°: Top: Total columns (in $10^{16}$ molec.cm$^{-2}$) of IASI observations (red) and the regression fit without the VPSC proxy (black), for a subset of the time series, zooming on denitrification periods. The correlation coefficients between the fit and the IASI data (R) are displayed, as well as the root mean square error (RMSE). Middle: Same as top panels but for the fit with the VPSC proxy. Bottom: Normalized VPSC proxy. Note the different time and value ranges between the two hemispheres.

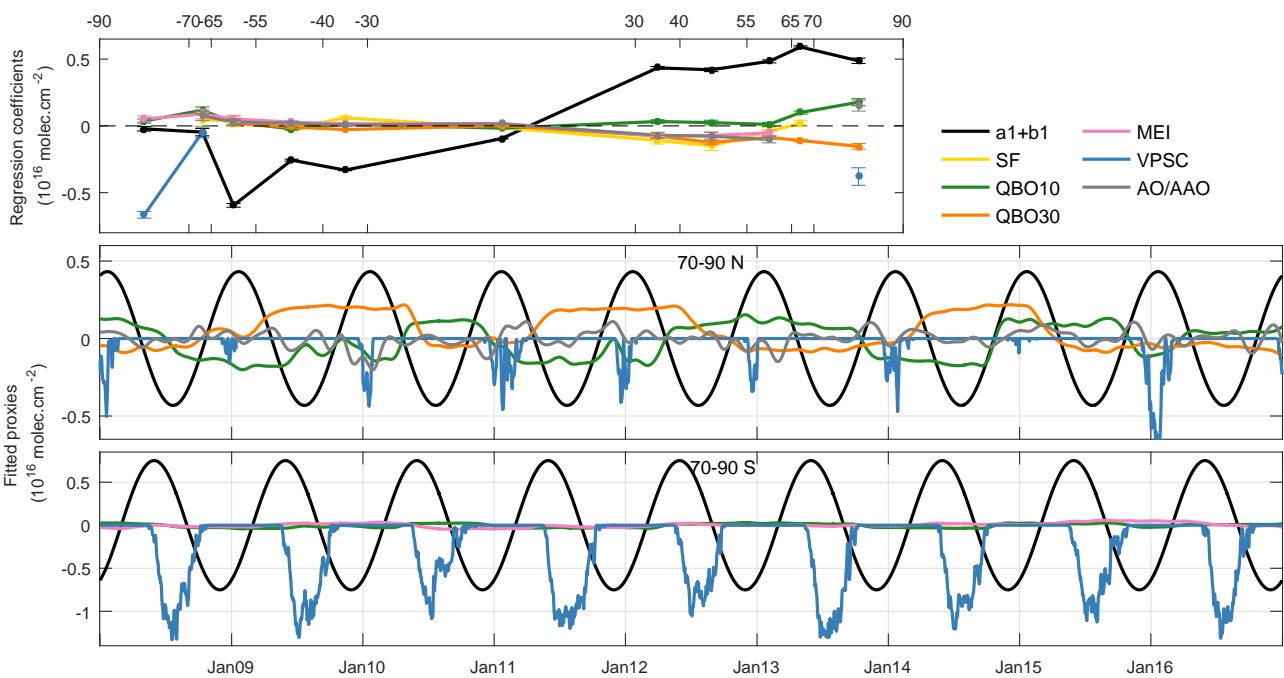

**Figure 8.** Top panel: Regression coefficients ($x_i$) and their standard error ($\sigma_e$, error bars, calculated by Eq. 3) for the selected variables in each equivalent latitude band (each data point is located in the middle of its corresponding eqlat band). Bottom panels: Fitted signal of the proxies in the eqlat bands 70-90 north (middle) and south (bottom) for the variables selected. They are calculated as [$x_i.X_i$] with $X_i$ the normalized proxy and $x_i$ the regression coefficient calculated by the regression model.

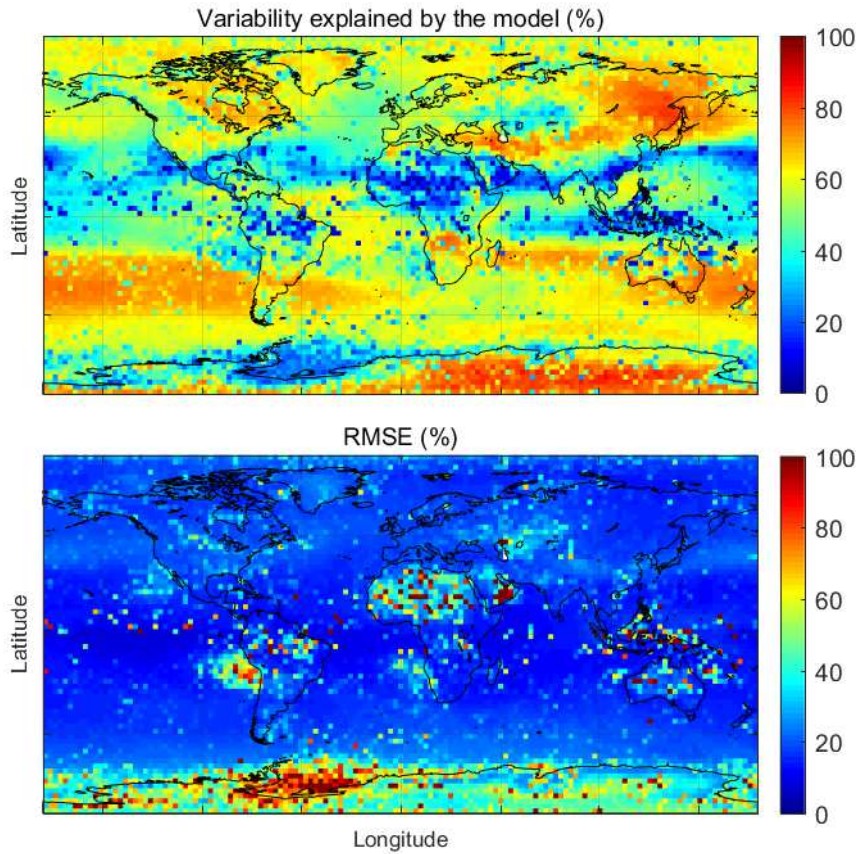

**Figure 9.** Top: Fraction (%) of the HNO$_3$ variability in the IASI observations explained by the regression model, and calculated as $[\sigma(HNO_3{}^{\text{fit}})/\sigma(HNO_3{}^{\text{IASI}}) \times 100]$. Bottom: Root Mean Square Error (RMSE) calculated for each grid cell as $[\sqrt{\dfrac{\sum(\text{fit-IASI})^2}{n}}]$ and expressed in %.

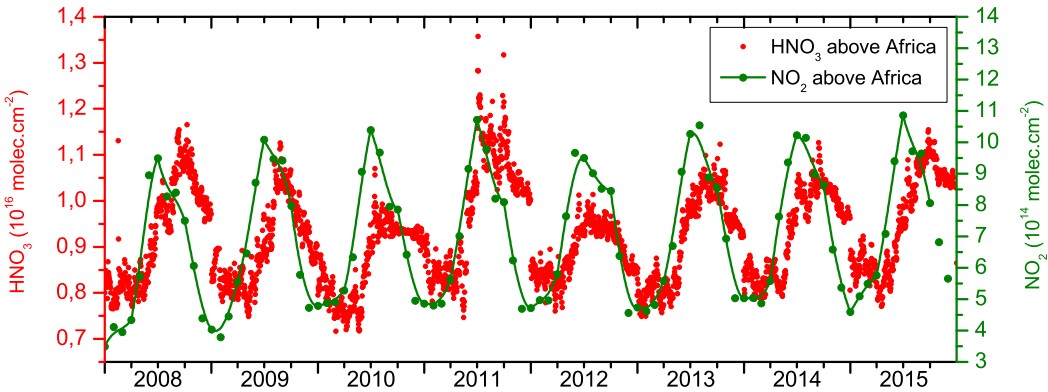

**Figure 10.** Time evolution of IASI $HNO_3$ (red) and GOME-2 $NO_2$ (green) from 2008 to 2015 for Africa south of the equator (5-20°S, 10-40°E). Both $HNO_3$ and $NO_2$ columns are expressed in molec.cm$^{-2}$. The $NO_2$ data are tropospheric columns (Valks et al., 2011) and are obtained from ftp://atmos.caf.dlr.de/. Note that the ranges differ between the two y-axes.

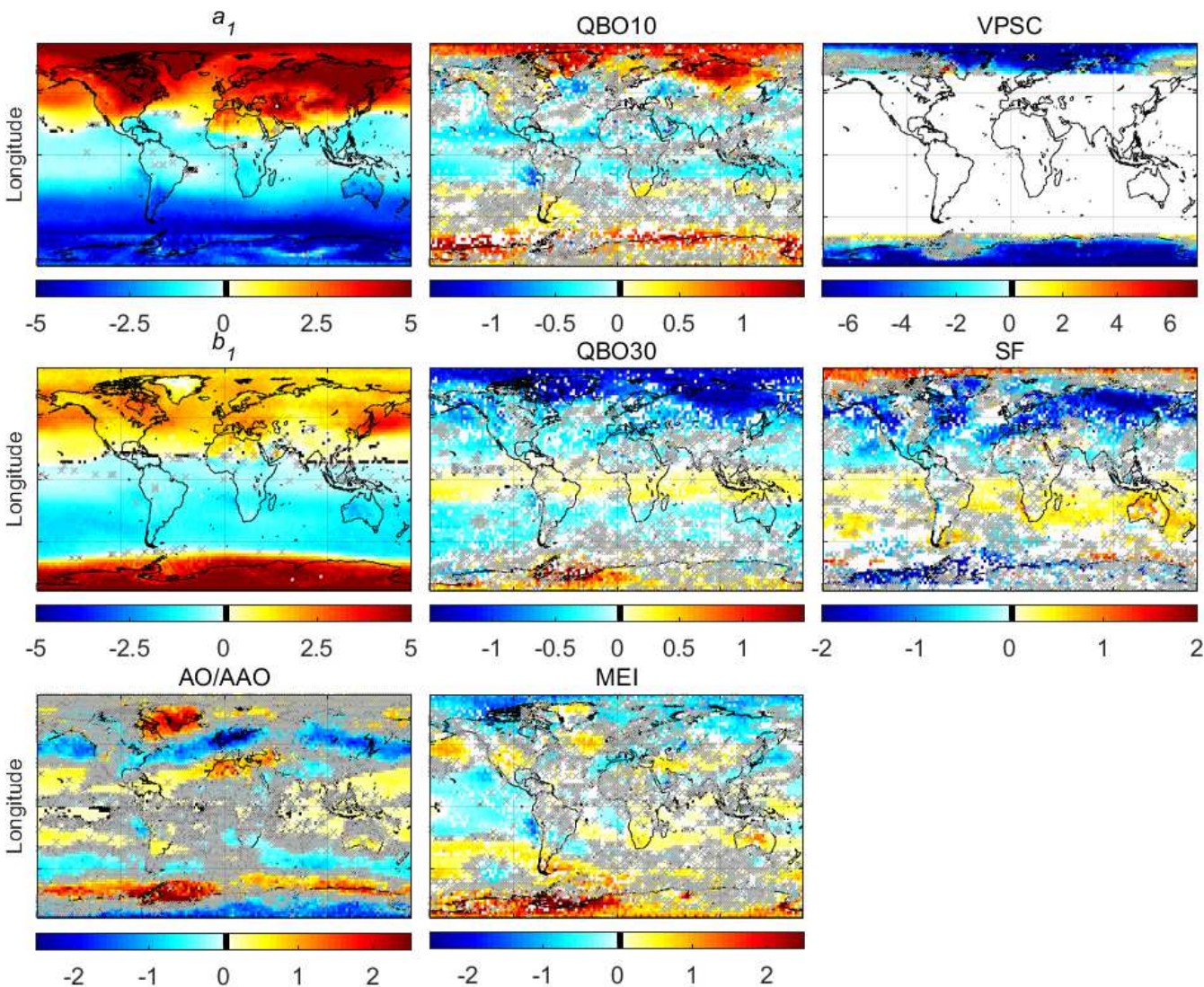

**Figure 11.** Global distributions ($2.5° \times 2.5°$ grid) of the regression coefficients expressed in $10^{15}$ molec.cm$^{-2}$. The gray crosses are the cells where the proxy is not significant when accounting for autocorrelation (see Eq.3). The white cells are where the proxy was not retained and the black cells represent a coefficient of 0. Note the different scales. The X axes are latitudes.