# Peer review of "Spatio-temporal variations of HNO3 total columns from 9 years of IASI measurements - A driver study"

_Atmospheric Chemistry and Physics, 2017_

## Referee Comment (RC1) · S. Solomon (Referee) · 31 Dec 2017

General comment: This paper presents and analyzes 9 years of total HNO3 column abundances from IASI measurements. The paper does a great job of presenting the data and discussing its key features. It will be a valuable resource for other researchers and merits publication. I do have a few comments and suggestions that I hope the authors will consider, presented below.

1) There is a lot of discussion of the apparent onset of denitrification and its association

[Figure]

with a 'threshold' T<195K. I think the paper would be stronger if this were accompanied by a discussion of the uncertainties in the temperatures used to identify this relationship (i.e., ERA-interim). Three questions arise a) Could it really be 194K within uncertainties? Why or why not? How sharp is this threshold within uncertainties? b) what about the influence of small scale waves that may not be resolved by the reanalysis? c) would there be value in simply doing a scatter plot of local HNO3 versus local temperature in winter, as opposed to the current approach of binning by eqlat?

2) Figures, 2, 3, and 4 are key results of this study, showing very well the seasonal cycles in the two hemispheres. A minor comment on Fig 2 is that heavy tics are needed for January, so one can see the exact mapping in time more readily. A substantive comment is that the time lag for recovery of the HNO3 column in the southern hemisphere is striking. The paper has some good discussion on this but I wonder if more could be done. In particular, if the mechanism replacing the HNO3 is mainly via transport, then HNO3 and O3 would show very similar post-final warming increases – do they? This could be shown, and would make the paper more useful. Further, I suspect that the very late recovery of HNO3 may have more to do with chemistry, in particular the fact that perpetual sunlight means NO3 is photolyzed effectively throughout the polar summer so that little of it can end up in N2O5 and hence to HNO3. This may do a better job explaining the ramp from Mar-May (e.g., in Fig 4) than other explanations, and could be probed fairly simply using chemical kinetic equations considering length of night, temperature, and ozone near 20-25 km. Zonal means should suffice, despite the potential for vortex meandering, which is likely to be small in summer.

3) The authors often (not always, but often) refer to low HNO3 columns as synonymous with denitrification, but what about, for example, high ClONO2 columns associated with the collar implying that NOy may be in that form at times? Similarly, the low HNO3 during mid-summer seen in Figure 4 and discussed above may have to do in part with perpetual sunlight meaning high NO and NO2 amounts, with less in HNO3 but not necessarily less total NOy. Some further discussion is merited.

---

## Referee Comment (RC2) · Anonymous Referee #2 · 3 Jan 2018

This paper presents 9 years of daily global total column HNO3 measurements from IASI and, for the first time, performs a detailed MLR analysis on them to statistically characterize the relative contributions of several explanatory variables in controlling the stratospheric HNO3 distribution. The manuscript is generally well organized and well written, and the production quality of the figures is also high. In my opinion, the study makes a valuable contribution and warrants publication. I do, however, have a number of mostly minor substantive comments (detailed below) that I would like to see addressed before the manuscript is accepted for publication.

General comment:

[Figure]

\* One general – though minor and easily rectified – comment is a pervasive lack of adequate referencing throughout the manuscript. PSC formation and denitrification, and their roles in chlorine activation and chemical ozone loss, are extremely well-studied phenomena, and obviously it is not possible (or even desirable) to cite every paper on these topics published in the last 30 years. But in many places the authors have chosen to cite only a few papers for well-known points, without prefacing the list with "e.g.". This may seem like a petty point, but not only does their selection of which papers to reference often come across as arbitrary, but also their approach may give non-expert readers the impression that only those few highlighted papers are of relevance. So I suggest going through the manuscript and adding "e.g." in front of the list of cited papers in many places. Some specific examples of where this is needed include: p2, L4; p2, L6; p2, L8; p2, L19; p2, L23; p8, L2; p8, L4; p8, L7; p8, L17; p8, L18; p8, L24. Similarly, although the source (typically a URL) for each proxy is given in Table 1, I feel that it would be appropriate to provide a general citation in each sub-section of Section 4.3 where a given proxy is introduced. For example, references to published literature are needed on p7, L27 for F10.7, p8, L14 for MEI, and p8, L21 for AO and AAO.

Specific substantive comments and questions:

\*p2, L26: I do not think it is true that "most often" MLR studies use an iterative selection procedure to identify relevant explanatory variables. In fact, I believe that only a handful of the many MLR ozone studies have done so. (And it seems strange to say "most often" and then cite only one reference.)

\*p3, L30: Does the cloud screening of IASI data include PSCs?

\*p4, L9: The PSC formation threshold is stated to be 195 K. It is fine for the purposes of this kind of analysis to use a constant value to indicate the likely presence of PSCs, but it should be acknowledged that the temperature at which NAT forms varies with altitude and time over the season, and thus this value is approximate.

\*p5, L4-5: It is true that these IASI results confirm earlier findings, and references are
needed here.

*p5, L9-11: I find this part of the discussion confusing. First, it is stated that the "delayed denitrification" in the 65-70S band is attributable to "the later appearance of PSCs" and "the mixing of these air masses with the denitrified air masses from the center of the vortex". Are the authors asserting that some of the decrease in HNO3 observed in the 65-70S band does not arise directly from PSC sedimentation within that band, but rather from dilution of HNO3 abundances through mixing with denitrified air masses from deeper in the vortex core? In that case, the decrease in HNO3 should not be called "denitrification". More importantly, is this suggestion consistent with the findings of Roscoe et al. [JGR 117, 2012] that the broad vortex edge region is only weakly mixed with the deep core during the winter? Second, the next sentence states that these "two processes lead to the total columns in both eqlat bands being in the same range of values by the end of December". The Antarctic vortex is breaking down (or has mostly broken down) by the end of December, so of course mixing at this time homogenizes the high-latitude HNO3 distribution, but it doesn't make sense to be talking about the later appearance of PSCs in this context.

*p5, L16-18: I also find these sentences confusing. It is stated that the columns in the 55-65S band keep increasing during the low-temperature periods, but cold intervals are not marked for that eqlat band. Are the authors referring to periods that are cold at higher latitudes? If so, then this statement is not entirely correct, as HNO3 values at 55-65S start to decline from their peak values while temperatures are still low in the 70-90S and 65-70S bands. The maximum in HNO3 values in June-July is attributed to "less sunlight compared to lower latitudes", but the comparison shouldn't be to lower latitudes but rather midwinter vs summer (at the same latitude). In addition, the role of confined diabatic descent inside the vortex should be mentioned, as it is a major factor leading to strongly enhanced wintertime HNO3 abundances in the lower stratospheric layer to which the IASI column amounts are most sensitive.

*p5, L23-24: The statement that temperatures in the northern high latitudes rarely

reach the PSC formation threshold is much too general. While that is true for the polar-cap (70-90N) average being considered here, temperatures in the Arctic lower stratosphere certainly do drop below PSC formation thresholds in localized regions in almost every year. Moreover, it is not the *average* temperature – which is what I believe is being shown in Figure 2, although it's not clear – that is important for PSC formation. It is the *minimum* temperature that is important. In fact, if indeed Figure 2 is showing eqlat band average temperature, then it should be reformulated to correlate HNO3 behavior with the minimum temperatures in that band. In any case, the exact nature of the temperatures being shown should be specified (at the beginning of Section 3 and in the caption).

*p6, L4-5: For ease of reference, the lack of IASI data in September-December 2010 should be first noted in Section 2, where the data set is described. It seems to me that this interval is also noticeable in Figure 2, so I suggest removing the data during this period in that Figure as well.

*p6, 7-9: It is hypothesized that the anomalous behavior in July-August 2010 seen in IASI HNO3 data was a consequence of descent induced by the midwinter minor warming. It seems to me that a more obvious explanation is that the SSW caused lower stratospheric temperatures to rise sufficiently that PSC formation was temporarily inhibited. It is worth noting in the manuscript that a similar evolution of HNO3 was recorded by Aura MLS in that winter, as shown in Figure 3-6 of the 2014 WMO Ozone Assessment. The 2014 WMO Report also showed that in 2010 VPSC (based on MERRA) remained well below the 1979-2012 Antarctic average and less denitrification than typical occurred.

*p10, L20-21: I find this discussion confusing. First, a *delay* in the drop in HNO3 concentration in the fit for the 65-70S band is noted, but then it is stated that it "happens *earlier* than in the IASI observations since the VPSC proxy is based on temperatures and composition *north* of 70". Figure 6 does show that the fitted midwinter peak in HNO3 slightly precedes that observed, so I assume that "earlier" is correct and "delay"

[Figure]

must be a typo. However, it is also true that the HNO3 decline is more gradual in the model than in the data, so that in late winter the fit line lags the observations. Exactly which behavior is being discussed should be clarified. Also, since it is the Antarctic that is being talked about here, "north" should be "poleward".

*p10, L27-28: The deep minima in HNO3 in the northern polar regions in October 2014 and 2016 almost certainly have nothing to do with denitrification during the preceding Arctic winters. Any signature of denitrification gets completely obliterated when the vortex breaks down at the end of winter. Even in the Antarctic, where denitrification is severe every winter, its signature is not still visible in the high-latitude HNO3 abundances the following fall. The extremely low 70-90N HNO3 values in October 2014 and 2016 (and also 2012, when the residuals are particularly large) are indeed quite interesting, but they cannot be ascribed to denitrification. It's possible that the low HNO3 observed in boreal fall 2016 may have been linked to the QBO disruption [e.g., Tweedy et al., 2017].

*p11, L14-15: It is noted that parts of Eurasia stand out with a low percentage of observed variability explained by the model. Could this be related to the low sensitivity of IASI data in this region, where the elevated terrain of the Tibetan Plateau reduces the signal-to-noise of the retrieval (e.g., Luo et al., ACPD 2017)?

*p11, L16-21: The low fraction of explained HNO3 variability in the tropics and subtropics is attributed to lightning NOx production. In addition to sources, unaccounted-for sinks of HNO3 should also be considered, such as scavenging in convective updrafts and cirrus clouds.

*p11, L27-28: If the signal over southern Africa induced by NO2 from biomass burning is being carried by the annual term in the model, then shouldn't the coefficients a1 or b1 be larger in that region in Figure 11 (which is not the case)?

*p12, L1-2: It might be good to mention the issues with the retrievals caused by elevated terrain here as well.

*p12-13, Section 4.4.3: I appreciate that the authors limited the number of figures, showing only the regression coefficient for each proxy (Figure 11) and not the fraction of HNO3 variability it explains. But the accompanying discussion frequently refers to the percentage contribution from specific proxies. Although some sense of their relative importance in different regions can be obtained from Figure 11 (and also Figure 8), I suggest either adding "(not shown)" everywhere a percentage contribution is discussed in this section or adding (and referring to) another figure containing this information.

*p12-13, Section 4.4.3: I would have liked to have seen a bit more discussion of whether these results for HNO3 are consistent with previous MLR analyses of ozone data that included similar terms. In particular, the SF results are not put into the context of previous findings. In addition, the positive signal above the southern polar region is characterized as "weak", but in fact the largest positive MEI regression coefficients are found over Antarctica. Is that in line with expectation? Previous studies looking at the influence of AO/AAO on ozone are alluded to on p13, L8, but no references are given there, and it is not clear whether the citations in the next sentence are relevant for this point (e.g., the 2009 paper by Wespes et al. is about HNO3 and does not discuss the AO/AAO). The influence of the QBO in the equatorial regions is noted, but no mention is made of the fact that the coefficients are much larger at northern high latitudes.

*p14, L32-33: I do not wish to take away from the value of the IASI HNO3 measurements, whose dense spatial coverage and long-term record are obviously of great benefit, as this study has shown. But I would ask for a bit more care in the language used here. Although the novel statistical nature of these results is mentioned, I think that some readers could take away from these lines the message that this analysis has revealed the profound influence of PSC formation and denitrification on the HNO3 distribution, when in fact the crucial role of those processes has been known for decades. In truth, it is not obvious to me what additional knowledge about the variability of HNO3 in the polar regions has been gained from this study that had not been demonstrated previously using limb measurements with much coarser horizontal but much greater

vertical resolution.

*p25, Figure 2: Minor tick marks on the y-axis would be helpful. As mentioned earlier, it might be good to remove the sparse measurements during the September-December 2010 interval from this plot as well. Why do some of the vertical lines, especially (but not only) in the purple 65-70 eqlat region, appear to be thicker? Is it because temperatures are hovering around the PSC threshold at those times, so the shading is being turned off and on multiple times in quick succession?

*p30, Figure 7: Why is there a break in the without-VPSC fit curve in the 70-90S panel in October-November 2014? Such a break does not appear in the similar panel for the with-VPSC fit (or in Figure 6).

*p33, Figure 10 caption: The wording of the caption ("Time evolution of IASI HNO3 (red) and NO2 (green)") implies that the NO2 data are from IASI, but the reference cited is for GOME-2 data. Please clarify.

Minor points of clarification, wording / figure suggestions, and grammar / typo corrections:

*p1, L13: "PSCs" should be defined in the abstract as well as the main body of the paper.

*p1, L23: inexsitent –> nonexistent

*p2, L3: "PSCs" was already defined on p1, L20

*p2, L8: and further –> followed by

*p2, L11, L14: These acronyms (UARS, MIPAS, ACE-FTS) should probably be spelled out. Also, "AURA" –> "Aura" and "ODIN" –> "Odin" (they are not acronyms, just names)

*p3, L10: bi-daily –> twice daily ("bi-daily" could be interpreted to mean every two days)

*p3, L15: The university name should be spelled out here

*p3, L24: Can 15-20 km really be considered the "low-middle" stratosphere? This seems more like just the lower stratosphere to me.

*p3, L30: higher fractional cloud cover than 25% –> fractional cloud cover higher than 25%

*p4, L19: Further than –> Beyond

*p4, L22: delete "columns" (some of the previous studies were based on HNO3 profiles, not columns)

*p5, L14: delete "itself"

*p5, L20: more –> longer

*p5, L26: It would be good to add "Arctic" in front of "winters" and "over a broader area" after "threshold"

*p5, L35: "polar" –> "potential"

*p6, L1: it's not clear why only one contour is noted here, when 3 contours of PV are shown in both hemispheres

*p6, L4: EUMETSAT should be in all capital letters (as on p3, L29)

*p6, L12: dentrification –> denitrification

*p6, L14: What does "more stable" mean in this context? More constant over the season, or more uniform from year to year?? And what is the comparison against – wintertime values in the NH, or summertime values in the SH?

*p6, L21: I assume that "Cst" in Eqn (1) is a constant term, but it should be defined

*p6, L29: Kyrola et al. [2010] seems like an odd reference for such a general statement about the BDC. Wouldn't the Butchart [2014] review paper (already cited elsewhere) be a better choice?

*p7, L13: further –> below

*p7, L19: ENSO should also be mentioned here

*p8, L13: meridonal –> meridional

*p8, L23: I assume it is meant that AO/AAO are considered only in the high latitudes of the hemisphere they are related to, since they are both applied in equatorial regions.

*p8, L29: delete "(from 195 K or TNAT for the formation of nitric acid trihydrate particles)"

*p8, L30: "PSCs" has already been defined

*p8, L31: delete "either"

*p9, L2: NAT has already been defined

*p9, L3: gaz –> gas

*p9, L13: dynamic –> dynamics

*p9, L23: Section 3.2 –> Section 4.2

*p9, L26: add "bands" after "90S"

*p9, L29: add "major" after "Most"

*p9, L30: RMSE is used here for the first time but not defined until p11, L31

*p10, L6: denitrifications –> denitrification seasons

*p10, L11: delete "here"

*p10, L17: better –> improved

*p10, L23: dynamic –> conditions

*p11, L5: The reference to Section 4.4.1 is incorrect (this part of the discussion is itself

in Section 4.4.1)

*p11, L11: most –> much; also add "generally" in front of "between"

*p11, L19: between –> in

*p11, L21: emitted –> produced

*p11, L25: oxydation –> oxidation

*p11, L34: desertic –> desert

*p12, L30: For MEI, "south of Africa" the results are not significant. I think "west of South Africa" would be better here.

*p13, L5: Groenland –> Greenland

*p13, L10: largely –> strongly

*p13, L21: further –> subsequent

*p13, L25: add "proxy" after "VPSC" and "HNO3" in front of "variability"

*p14, L7: reveal –> reflect

*p14, L14: between –> in

*p14, L23: but still allow improving significantly the model-to-observation agreement –> but accounting for PSCs still significantly improves the model-to-observation agreement

*p29, Figure 6: The font for the year labels on the x-axis seems to be disproportionately large

---

## Author Comment (AC1) · 1 Feb 2018

**Response to S. Solomon**

First, we would like to thank Prof. Solomon for her insightful comments and suggestions. They have all been taken into account and they have allowed improving the paper. Each of the comments is addressed on a point-by-point basis hereafter, with the original comment reproduced in blue first.

1) There is a lot of discussion of the apparent onset of denitrification and its association with a 'threshold' T<195K. I think the paper would be stronger if this were accompanied by a discussion of the uncertainties in the temperatures used to identify this relationship (i.e., ERA-interim). Three questions arise a) Could it really be 194K within uncertainties? Why or why not? How sharp is this threshold within uncertainties? b) what about the influence of small scale waves that may not be resolved by the reanalysis? c) would there be value in simply doing a scatter plot of local HNO3 versus local temperature in winter, as opposed to the current approach of binning by eglat?

We agree with the referee that the temperature of 195 K for the formation of PSCs can vary, namely depending on the conditions of the atmosphere, and that small scale waves would likely influence the local formation of PSCs. We plotted, as suggested by the referee, the HNO3 values versus the temperature for each grid cell between 70 and 90°S of equivalent latitude for the winter months (May to October, see Figure here below). The appearance of low HNO3 values occurs as expected around the 195 K threshold (red line), albeit with some variability, namely between 190 and 200 K. These variations are of interest when studying the precise dynamics of PSC formation and the results below suggest that IASI could make a strong contribution in monitoring the processes at play. We thank the referee for having triggered us to look into this in more details; we believe a forthcoming work could be undertaken regarding this question. However, for the present study, we believe that an average temperature of 195 K is a reasonable value to work with. We have mentioned these potential temperature variations in the text: "It should be noted that while this temperature is a widely accepted approximation for the formation threshold for NAT (type I), its actual value can be different depending on the local conditions (Lowe and MacKenzie, 2008; Drdla and Müller, 2010; Hoyle et al., 2013)." (p. 4, line 16-18)

---

## Author Comment (AC2) · 1 Feb 2018

Response to Referee #2

First, we would like to thank Referee #2 for his/her careful and expert reading of the paper as well as for his/her questions and suggestions. We have taken all the comments into account and have tried to address each as thoroughly as possible. We believe that the paper has been substantially improved thanks to the review.
The reviewer's comments are addressed here below on a point-by-point basis, with the initial comment reproduced in blue first.

General comment:
1) One general – though minor and easily rectified – comment is a pervasive lack of adequate referencing throughout the manuscript. PSC formation and denitrification, and their roles in chlorine activation and chemical ozone loss, are extremely well-studied phenomena, and obviously it is not possible (or even desirable) to cite every paper on these topics published in the last 30 years. But in many places the authors have chosen to cite only a few papers for well-known points, without prefacing the list with "e.g.". This may seem like a petty point, but not only does their selection of which papers to reference often come across as arbitrary, but also their approach may give non-expert readers the impression that only those few highlighted papers are of relevance. So I suggest going through the manuscript and adding "e.g." in front of the list of cited papers in many places. Some specific examples of where this is needed include: p2, L4; p2, L6; p2, L8; p2, L19; p2, L23; p8, L2; p8, L4; p8, L7; p8, L17; p8, L18; p8, L24.

The mention "e.g." was added everywhere as was suggested by the reviewer. A few additional references were also included to be more exhaustive, for example:

- p. 1 line 20
- p. 4 line 19
- p. 4 line 31
- p. 5 line 19
- p. 8 line 28
- p. 9 line 31
- p. 14 line 34
- p. 15 line 10

Similarly, although the source (typically a URL) for each proxy is given in Table 1, I feel that it would be appropriate to provide a general citation in each sub-section of Section 4.3 where a given proxy is introduced. For example, references to published literature are needed on p7, L27 for F10.7, p8, L14 for MEI, and p8, L21 for AO and AAO.

General references were added for the description of the proxies:
- For the solar flux: *"…,and correlates to the number of sunspots on the solar disk (Covington, 1948; Tapping and DeTracey, 1990; Tapping, 2013)."* (p. 8 line 24-25)
- For the MEI: *"…, surface air temperature and cloudiness fraction (Wolter and Timlin, 1993, 1998)."* (p. 9 line 14)
- For the AO/AAO: *"…in the northern and southern hemispheres, respectively (Gong and Wang, 1999; Kodera and Kuroda, 2000; Thompson and Wallace, 2000)."* (p.9 line 20-21)

Specific substantive comments and questions:
2) p2, L26: I do not think it is true that "most often" MLR studies use an iterative selection procedure to identify relevant explanatory variables. In fact, I believe that only a handful of the many MLR ozone studies have done so. (And it seems strange to say "most often" and then cite only one reference.)

We agree with this remark. We have removed the "most often" and changed the sentence to "*In various multivariate regression studies, an iterative selection procedure is used to isolate the relevant variables for the concerned species*" (p. 2 line 26-27). Regarding the reference, it was initially meant to lead the reader to a paper describing that method (Mäder et al., 2007). However, for more completeness, other references have also been added: (Steinbrecht, 2004; Mäder et al., 2007; Knibbe et al., 2014; Wespes et al., 2016, 2017). (p. 2 line 26-28)

**3)** p3, L30: Does the cloud screening of IASI data include PSCs?

The cloudiness in the IASI pixel is estimated from the Advanced Very-High-Resolution Radiometer (AVHRR) satellite imager. The AVHRR observations are mapped inside the IASI pixel to determine the mean percentage of cloud cover in each IASI pixel. A maximum threshold of 25% for the cloud cover has been chosen for considering the IASI pixel as clear for the $HNO_3$ retrievals. While thick cirrus clouds have detectable signatures in the IASI spectra (on the longwave part of the spectrum), we have not been able to see any feature in the presence of PSCs, which are presumably absorbing IR radiation much too weakly. With this in mind, it is safe to say that scenes with PSCs are not flagged as cloudy and will not be filtered out by the operational processing for the $HNO_3$ retrievals.

Note also that the PSCs have, to the best of our knowledge, never been detected from the nadir spectra. They have already been measured from infrared limb sounders such as MLS, MIPAS (e.g. Höpfner et al. 2006, 2009; Nakajima et al., 2016) which have a longer observing path.

**4)** p4, L9: The PSC formation threshold is stated to be 195 K. It is fine for the purposes of this kind of analysis to use a constant value to indicate the likely presence of PSCs, but it should be acknowledged that the temperature at which NAT forms varies with altitude and time over the season, and thus this value is approximate.

We thank the referee for making this point, on which we fully agree. The figure below gives $HNO_3$-temperature correlations for the stratosphere and it is indeed seen that low $HNO_3$ columns occur also at lower or higher temperatures than 195 K (namely between 190 and 200 K), on a local scale. This section was adapted to clarify this and put more caution on the 195 K threshold: *"… (195 K, based on ECMWF temperatures). It should be noted that while this temperature is a now widely accepted approximation for the formation threshold for NAT (type I), its actual value can be different depending on the local conditions (Lowe and MacKenzie, 2008; Drdla and Müller, 2010; Hoyle et al., 2013). Also, other forms of PSCs, particularly the type II PSCs (ice clouds) form at a lower temperature of 188 K, corresponding to the frostpoint of water, or 2-3 K below that (e.g. Toon et al. (1989); Peter (1997); Tabazadeh et al. (1997); Finlayson-Pitts and Pitts (2000))."* (p. 4 line 16-20)

[Figure]

Figure 1. Total columns of $HNO_3$ ($10^{16}$ molec.cm$^{-2}$) VS temperatures (K) for the 70-90 S eqlat band and between May and October. The color scale indicates the number of measurements per bin (with grey areas for bins with less than 80 measurements), and the red line is the 195 K threshold.

5) p5, L4-5: It is true that these IASI results confirm earlier findings, and references are needed here.

We had references for this further in the text, but we agree that they are needed here too. The references (McDonald et al., 2000; Santee et al., 2004; Lambert et al., 2016) have been added. (p. 5 line 15)

6) p5, L9-11: I find this part of the discussion confusing. First, it is stated that the "delayed denitrification" in the 65-70S band is attributable to "the later appearance of PSCs" and "the mixing of these air masses with the denitrified air masses from the center of the vortex". Are the authors asserting that some of the decrease in HNO3 observed in the 65-70S band does not arise directly from PSC sedimentation within that band, but rather from dilution of HNO3 abundances through mixing with denitrified air masses from deeper in the vortex core? In that case, the decrease in HNO3 should not be called "denitrification". More importantly, is this suggestion consistent with the findings of Roscoe et al. [JGR 117, 2012] that the broad vortex edge region is only weakly mixed with the deep core during the winter?

We would like to thank the referee for this comment and the reference mentioned. Indeed, we had initially attributed the delay in denitrification to the mixing of air masses between the two regions of the vortex in addition to the PSC sedimentation (the purple shaded areas in Fig.2 indicate the threshold temperature for the PSC Ia formation). It is however clear, following the above reference and others, that this process is unlikely to contribute significantly. The text was accordingly changed to: "*The delayed decrease in $HNO_3$ in the outer parts of the vortex (i.e. in the 65-70 S eqlat band) can thus be attributed to the later appearance of PSCs in this region (see Figure 2 purple shaded areas in second panel)."* (p. 5 line 17-18)

Also, mention of the separation between the two parts of the vortex is made further in the text: "*It is worth noting that the two regions previously mentioned (inner and outer vortex) have been observed to behave differently; the inner vortex (70-90 S) undergoes strong internal mixing whereas the outer vortex (65-70 S), isolated from the vortex core, experiences little mixing of air. This, combined to a cooling of the stratosphere, could lead to increased PSC formation and further ozone depletion (Lee et al., 2001; Roscoe et al., 2012)."* (p. 5 line 35 – p. 6 line 3)

Second, the next sentence states that these "two processes lead to the total columns in both eqlat bands being in the same range of values by the end of December". The Antarctic vortex is breaking down (or has mostly broken down) by the end of December, so of course mixing at this time homogenizes the high-latitude HNO3 distribution, but it doesn't make sense to be talking about the later appearance of PSCs in this context.

Following the changes made with regard to the previous point, this sentence was also modified to: "*By the end of December, i.e. when the vortex has started breaking down (e.g. Schoeberl and Hartmann, 1991; Manney et al.,1999; Mohanakumar, 2008), the total columns in both eqlat bands homogenize and reach the same range of values $(1.7x10^{16}molec.cm^{-2})$."* (p. 5 line 19-21)

7) p5, L16-18: I also find these sentences confusing. It is stated that the columns in the 55-65S band keep increasing during the low-temperature periods, but cold intervals are not marked for that eqlat band. Are the authors referring to periods that are cold at higher latitudes? If so, then this statement is not entirely correct, as HNO3 values at 55-65S start to decline from their peak values while temperatures are still low in the 70-90S and 65-70S bands. The maximum in HNO3 values in June-July is attributed to "less sunlight compared to lower latitudes", but the comparison shouldn't be to lower latitudes but rather midwinter vs summer (at the same latitude).

With this sentence, we meant that the $HNO_3$ columns in that eqlat band do not drop as suddenly as at higher latitudes, with regard to the start of the low temperatures at higher

latitudes. We agree that the formulation was not sufficiently clear and the sentence was therefore changed to: "*…, we show that the columns in that band keep increasing when the temperatures at higher latitudes start decreasing, to reach maximum values of about 3.4x10$^{16}$ molec.cm$^{-2}$ in June-July; this is due to a change in the NO$_y$ partitioning towards HNO$_3$, itself due to less sunlight compared to the summer.*" (p. 6 line 6-8)

> In addition, the role of confined diabatic descent inside the vortex should be mentioned, as it is a major factor leading to strongly enhanced wintertime HNO3 abundances in the lower stratospheric layer to which the IASI column amounts are most sensitive.

The role of this diabatic descent was mentioned in the part analysing the annual cycle in Section 4.4.3 (p. 13 line 25-29). However, we agree with the referee that it would be good to include it here first, since we indeed already describe the higher columns recorded during the winter. This section thus now reads: "*Also inducing increased concentrations during the winter at high latitudes is the diabatic descent occurring inside the vortex when the temperatures decrease. This downward motion of air enriches the lower stratosphere in HNO$_3$ coming from higher altitude (Manney et al., 1994; Santee et al., 1999), yielding higher column values which are, in this eqlat band, not affected by denitrification.*" (p. 6 line 9-12)

> **8)** p5, L23-24: The statement that temperatures in the northern high latitudes rarely reach the PSC formation threshold is much too general. While that is true for the polar-cap (70-90N) average being considered here, temperatures in the Arctic lower stratosphere certainly do drop below PSC formation thresholds in localized regions in almost every year. Moreover, it is not the *average* temperature – which is what I believe is being shown in Figure 2, although it's not clear – that is important for PSC formation. It is the *minimum* temperature that is important. In fact, if indeed Figure 2 is showing eqlat band average temperature, then it should be reformulated to correlate HNO3 behavior with the minimum temperatures in that band. In any case, the exact nature of the temperatures being shown should be specified (at the beginning of Section 3 and in the caption).

We agree with this general statement that temperatures in the northern hemisphere can locally reach below 195 K. However, we feel that for the purpose of the paper and the analysis, we should stick to average temperatures (what is shown here). Indeed, it makes more sense to us to talk about average temperatures when we treat average HNO$_3$ total columns. However, for the sake of completeness, we show the minimum temperatures in the Figure below. While local minima are indeed observed in the minimum temperatures (last panel), the general pattern of temperatures below 195 K is the same as when considering the average temperatures (5$^{th}$ panel). For this reason, and because we had rather stay consistent with the average HNO$_3$ total columns, we decided to keep the average temperatures. The legend of Figure 2 in the paper was completed: "***Figure 2**. (four top panels) HNO$_3$ total columns time series for the years 2008-2016, for equivalent latitude bands 70-90, 65-70, 55-65 and 40-55, north (green) and south (blue). Vertical shaded areas are the periods during which the average temperatures are below T$_{NAT}$ in the north (green) and south (blue) 70-90° band, and in the south (purple) 65-70° band. Note that the large period without data in 2010 is when there was a low amount of data distributed by EUMETSAT (see Section 2). (bottom panel) Daily average temperatures time series (in K) taken at the altitude of 50 hPa for the equivalent latitude bands 70-90° North (green) and South (blue) and 65-70° South (purple). The horizontal black line represents T$_{NAT}$ , i.e. the 195 K line.*" (p. 29)

Note also that mention of this was added in the text: "*The northern hemisphere high latitudes usually do not experience denitrification, mostly because the temperatures, while sometimes showing local minima below 195 K, rarely reach the PSC formation threshold on broad areas and for long time spans (see Figure 2 for average temperatures, light green vertical areas). A few years stand out, however, …*" (p. 6 line 16-18)

[Figure]

Figure 2. As Figure 2 in the text (see reproduced legend between quotes above). The 6th panel is the minimum temperatures time series (in K) for eqlat bands 70-90 N (green), 70-90 S (blue) and 65-70 S (purple).

**9)** p6, L4-5: For ease of reference, the lack of IASI data in September-December 2010 should be first noted in Section 2, where the data set is described. It seems to me that this interval is also noticeable in Figure 2, so I suggest removing the data during this period in that Figure as well.

We thank the referee for this remark. The explanation for this lack of data is now mentioned in Section 2, when describing the IASI data. The sentence was transferred from one section to the other, and is now: "*…, i.e. all scenes with a higher fractional cloud cover than 25% are not taken into account. It should be noted that there was an abnormally small amount of IASI L2 data distributed by EUMETSAT between the 14th of September and the 2nd of December 2010 (Van Damme et al., 2017), and that these data have been removed from the figures and analyses in the following of the paper. For the present study,…*" (p. 3 line 32 – p.4 line 3)

The part in Section 3 reads in turn: "*… in the northern (top) and the southern (bottom) hemispheres. July and August of 2010 stand out in the Antarctic, with high and variable columns…*" (p. 6 line 32-33)

Figure 2 was updated by removing these data. (p. 28)

**10)** p6, 7-9: It is hypothesized that the anomalous behavior in July-August 2010 seen in IASI HNO3 data was a consequence of descent induced by the midwinter minor warming. It seems to me that a more obvious explanation is that the SSW caused lower stratospheric temperatures to rise sufficiently that PSC formation was temporarily inhibited. It is worth noting in the manuscript that a similar evolution of HNO3 was recorded by Aura MLS in that winter, as shown in Figure 3-6 of the 2014 WMO Ozone Assessment. The 2014 WMO Report also showed that in 2010 VPSC (based on MERRA) remained well below the 1979-2012 Antarctic average and less denitrification than typical occurred.

Indeed, we conducted further research on the topic, thanks to the reference mentioned in this comment, and adapted this section accordingly: "*…, with high and variable columns*

*recorded by IASI. This is a consequence of a mid-winter (mid-July) minor sudden stratospheric warming (SSW) event that induced a downward motion of air masses and modified the chemical composition of the atmosphere between 10 and 50 hPa and until at least September (de Laat and van Weele, 2011). This is a consequence of a mid-winter (mid-July) minor sudden stratospheric warming (SSW) event that induced a downward motion of air masses and modified the chemical composition of the atmosphere between 10 and 50 hPa and until at least September (de Laat and van Weele, 2011; Klekociuk et al., 2011). The principal effect of this sudden stratospheric warming was to reduce the formation of PSCs (which stayed well below the 1979-2012 average, WMO (2014)) and hence reduce denitrification. This is shown by an initial drop in June, as is usually observed in other years but then by an increase in $HNO_3$ columns when the SSW occurs. These results confirm those previously obtained by the Aura MLS during that winter and reported in the World Meteorological Organization (WMO) Ozone Assessment of 2014 (see Figure 6-3, WMO (2014)). Apart from these peculiarities for the year 2010,…"* (p. 6 line 33 – p. 7 line 5)

11) p10, L20-21: I find this discussion confusing. First, a *delay* in the drop in HNO3 concentration in the fit for the 65-70S band is noted, but then it is stated that it "happens *earlier* than in the IASI observations since the VPSC proxy is based on temperatures and composition *north* of 70". Figure 6 does show that the fitted midwinter peak in HNO3 slightly precedes that observed, so I assume that "earlier" is correct and "delay" must be a typo. However, it is also true that the HNO3 decline is more gradual in the model than in the data, so that in late winter the fit line lags the observations. Exactly which behavior is being discussed should be clarified. Also, since it is the Antarctic that is being talked about here, "north" should be "poleward".

We apologize for the confusion in this paragraph. What is meant here is that, in the 65-70 S eqlat band, the fit calculated by the model is lagged in time (happens slightly earlier) compared to the observations in that band, because the VPSC proxy is based on the temperatures of the 70-90 S eqlat band, where the drop happens earlier. Hence the fit in the 65-70 S band simulates a drop earlier than is actually observed in that band. We hope this makes the understanding of this section easier, and decided, as suggested, to clarify it in the manuscript to avoid any future confusion. This section is now: "*This translates to a lag between the observations in the 65-70 S eqlat band and the fit in which the drop of $HNO_3$ concentrations happens earlier than in the IASI observations. This is explained by the fact that the VPSC proxy is based on temperatures and composition poleward of 70°. It induces a lower correlation coefficient (0.87) and…*". (p. 11 line 21-23)

12) p10, L27-28: The deep minima in HNO3 in the northern polar regions in October 2014 and 2016 almost certainly have nothing to do with denitrification during the preceding Arctic winters. Any signature of denitrification gets completely obliterated when the vortex breaks down at the end of winter. Even in the Antarctic, where denitrification is severe every winter, its signature is not still visible in the high-latitude HNO3 abundances the following fall. The extremely low 70-90N HNO3 values in October 2014 and 2016 (and also 2012, when the residuals are particularly large) are indeed quite interesting, but they cannot be ascribed to denitrification. It's possible that the low HNO3 observed in boreal fall 2016 may have been linked to the QBO disruption [e.g., Tweedy et al., 2017].

We thank the referee for this remark. It was not our intention to imply a causality link between the exceptional Arctic denitrification events and the unusual lows recorded in the $HNO_3$ columns. To clarify this, we changed the formulation to: "*In the same way, a few pronounced lows recorded by IASI, especially those in the Northern polar regions (mid-June to early October 2014 and 2016, for instance) are not captured by the model.*" (p. 11 line 28-30)

As for the QBO disruption: we agree that it could have an effect on the $HNO_3$ behaviour in polar regions, but it is not suggested by the model, where the inclusion of the QBO proxy does not improve the model/observations misfit in 2016. To asses this further and because

this disruption occurred around 40 hPa (e.g. Newman et al., 2016, Tweedy et al., 2017), we tested the inclusion of the QBO at 50 and 20 hPa (rather than a 30 and 10 hPa). This yields similar results (see Figure below) for all years – including 2016 – and therefore suggests that QBO is not the driving effect for the deep minima.

[Figure]

Figure 3. IASI HNO$_3$ total columns (red dots) for the 2 northernmost eqlat bands and the associated fitted model (black curves) featuring the QBO at 20 and 50 hPa.

**13)** p11, L14-15: It is noted that parts of Eurasia stand out with a low percentage of observed variability explained by the model. Could this be related to the low sensitivity of IASI data in this region, where the elevated terrain of the Tibetan Plateau reduces the signal-to-noise of the retrieval (e.g., Luo et al., ACPD 2017)?

We believe there might be some confusion here, as to what area was meant: the low percentage of explained variability that we are mentioning here refers to the area above Kazakhstan and the west Siberian plains, which are rather low in altitude (compared to the Tibetan Plateau, anyway). We have added this in the text to clarify the point: *"…although some continental areas (Northern part of inner Eurasia above Kazakhstan and the west Siberian plains) stand out with percentages..."* (p. 12 line 16-17)

Regarding the low sensitivity of IASI that the reviewer is referring to, it is a feature specific to tropospheric species such as CO.

Regarding HNO$_3$ columns, since the sensitivity of IASI is mostly in the stratosphere, the retrievals are barely affected by topography. Furthermore, retrievals with weak signal-to-noise ratio and small degree of freedom for signal would translate to a weak detected variability in IASI HNO$_3$, but not to a weak explained variability by the regression model.

**14)** p11, L16-21: The low fraction of explained HNO3 variability in the tropics and subtropics is attributed to lightning NOx production. In addition to sources, unaccounted-for sinks of HNO3 should also be considered, such as scavenging in convective updrafts and cirrus clouds.

We thank the referee for this remark. Indeed, some HNO$_3$ sinks are most probably also unaccounted for, and mention of this was added in the text: *"... missing some of the variability recorded in the observational data. Another cause for the discrepancies between the observations and the model could be unaccounted sinks of HNO$_3$, such as deposition in the liquid or solid phase and scavenging by rain. It should be noted that a small area of high explained variability is observed in Africa, …"* (p. 12 line 24-27)

**15)** p11, L27-28: If the signal over southern Africa induced by NO2 from biomass burning is being carried by the annual term in the model, then shouldn't the coefficients a1 or b1 be larger in that region in Figure 11 (which is not the case)?

Indeed, it should, and it actually is, but to a lower extent than the signal elsewhere on the globe. Because of the color scale chosen for the $a_1$ and $b_1$ distributions, it does not appear clearly. The color scale was chosen to avoid complete saturation in the northern latitudes. Here the same figure is reproduced with a different color scale, where the largest (negative) influence of the $a_1$ coefficient appears more frankly in southern Africa (see top left panel):

[Figure]

Figure 4. Global distributions (2.5° x 2.5° grid) of the regression coefficients expressed in $10^{15}$ molec. cm$^{-2}$. The contrast of $a_1$ and $b_1$ were modified in order to enhance the negative signal above south Africa.

We added mention of this in the text: "*Indeed, the large vegetation fires of Africa every year around July emit the largest amounts of $NO_x$ (compared to large fires of South America, Australia and southeast Asia). Their influence translates to an overrepresentation of the annual term (up to $-2x10^{15}$ molec.cm$^{-2}$) in the fitted model (although not clearly visible in Figure 11 because of the color scale chosen). This larger contribution of the annual variability thus yields…*". (p. 12 line 30-33)

**16)** p12, L1-2: It might be good to mention the issues with the retrievals caused by elevated terrain here as well.

See also answer to comment 13: The topography does not cause any particular problem for the retrieval of HNO$_3$. The degree of freedom for signal and retrieval total errors are only slightly smaller and larger, respectively, above elevated surface. This is more the case in the tropical regions where the tropospheric HNO$_3$ represents a large part of the total HNO$_3$ (e.g. Ronsmans et al., 2016).

**17)** p12-13, Section 4.4.3: I appreciate that the authors limited the number of figures, showing only the regression coefficient for each proxy (Figure 11) and not the fraction of HNO3 variability it explains. But the accompanying discussion frequently refers to the percentage contribution from specific proxies. Although some sense of their relative importance in different regions can be obtained from Figure 11 (and also Figure 8), I suggest either adding "(not shown)" everywhere a percentage contribution is discussed in this section or adding (and referring to) another figure containing this information.

Indeed, while we asked ourselves the question of a supplementary figure a few times, we decided not to add it in order to focus the reader on the essential figures. The percentages we use in the discussion of the distributions of the coefficients are obtained from the formula described in the 'introductory' part of that section ($[\sigma(X_i)/\sigma(HNO_3^{IASI}) \times 100]$). We do agree that the source of those percentages may be forgotten as one reads on through the manuscript. In order to keep the text as understandable as possible, we have chosen to add a sentence in the introductory part to insist on the use of the equation: *"…, and expressed in %. Note that, although the distributions of the contribution of each proxy are not shown as a Figure, the calculated percentage values are used in the following discussion (next 3 sections) to quantify the influence of the fitted parameters."* (p. 13 line 11-13)

**18)** p12-13, Section 4.4.3: I would have liked to have seen a bit more discussion of whether these results for HNO3 are consistent with previous MLR analyses of ozone data that included similar terms. In particular, the SF results are not put into the context of previous findings.

The paragraph was adapted in order to put our results in parallel with previous similar studies that addressed the question of the influence of the solar cycle on ozone. It was found that the behaviour of $HNO_3$ is mostly different from that of $O_3$ in most studies, in that we obtain a negative signal in the northern hemisphere. Mention of this is provided in the text as follows: *"… or negative elsewhere. Previous studies showed that ozone changes due to the solar cycle are largest in the low stratosphere (Hood, 1997; Soukharev and Hood, 2006), which corresponds to the altitude of maximum sensitivity for $HNO_3$. Our results for the mid to high latitudes suggest opposite behaviour for $HNO_3$ (as was also reported for $O_3$ by Wespes et al. (2017)). However, the positive contribution of the solar cycle on the $HNO_3$ variation in the tropical stratosphere is in line with the low-latitude $O_3$ response previously reported (Soukharev and Hood, 2006; McCormack et al., 2007; Frossard et al., 2013; Maycock et al., 2016)."* (p14 line 5-10)

In addition, the positive signal above the southern polar region is characterized as "weak", but in fact the largest positive MEI regression coefficients are found over Antarctica. Is that in line with expectation?

First, we would like to notify the reviewer that we did as for the solar cycle; we added references of previous studies in the text: *"…, and in the mid-latitudes of the northern hemisphere. The east-west gradient is in good agreement with chemical and dynamical effects of El Niño on $O_3$, and with previous studies that showed the same patterns for the influence of the MEI on $O_3$ (Hood et al., 2010; Rieder et al., 2013, Wespes et al., 2017)."* (p. 14 line 17-19)

As for the strong positive signal over west Antarctica, general caution should be taken when it comes to the results in that particular area of the southern polar regions. This region often shows unexpected retrieved concentration profiles for $HNO_3$ (see also the RMSE distribution in Figure 9, bottom), that we interpret as erroneous. This is likely related to emissivity issues in that region (mentioned in Section 4.4.2, p. 13 line 2-6) and in particular the fact that ice-shelf is treated with a constant ocean-like emissivity.

To avoid misinterpretation of this, we have added the following when describing the distribution of the coefficients: *"Note also that the strong negative signal observed above western Antarctica is most probably due to the drawback of using for all seasons a constant*

*emissivity for ocean surfaces (e.g. even when the ocean becomes frozen). For this reason, the regression coefficients in this area will not be discussed further."* (p. 14 line 10-13)

> Previous studies looking at the influence of AO/AAO on ozone are alluded to on p13, L8, but no references are given there, and it is not clear whether the citations in the next sentence are relevant for this point (e.g., the 2009 paper by Wespes et al. is about HNO3 and does not discuss the AO/AAO).

All references were listed at the end of this paragraph, but we agree that it is confusing. In the revised manuscript, we have divided the references between those for similar MLR studies and those detailing the influence of the AO/AAO in the atmosphere in general. Also, the reference of Wespes et al. 2009 was a typo, we meant Wespes et al. 2017. We thank the referee for his careful reading and apologize for this. The paragraph now reads: *"These results are in agreement with previous studies that showed that, for $O_3$, both the arctic and antarctic oscillations (also called "annular modes") are leading modes of variation in the extratropical atmosphere (Weiss et al., 2001; Frossard et al., 2013; de Laat et al., 2015; Wespes et al., 2017).They largely influence the circulation up to the lower stratosphere and represent, particularly in the southern hemisphere, the fluctuations in the strength of the polar vortex (Thompson and Wallace, 2000; Jones and Widmann, 2004; van den Broeke and van Lipzig, 2004). This further shows the similarity in the behaviour of $O_3$ and $HNO_3$."* (p. 14 line 24-29)

> The influence of the QBO in the equatorial regions is noted, but no mention is made of the fact that the coefficients are much larger at northern high latitudes.

Indeed, this point needed further explanation. This section in the text was completed in order to describe the potential influence of the QBO in the extratropics: *"Even though the QBO is a tropical phenomenon, its effects extend as far as the polar latitudes, through the modulation of the planetary Rossby waves (Holton and Tan, 1980; Baldwin et al., 2001). Because there are more topographical features in the northern hemisphere than in the southern hemisphere, these waves have a larger amplitude and can influence the polar stratospheric temperatures and hence the vortex formation. While the exact mechanism for the extratropical influence of the QBO is not exactly understood (Garfinkel et al., 2012; Solomon et al., 2014), it seems the large positive and negative signals observed in the northern high latitudes in Figure 11 can indeed be attributed to this modulation of the Rossby waves by the oscillation in the meridional circulation. This was also observed for $O_3$ by e.g. Wespes et al., (2017)."* (p. 14 line 33 – p. 15 line 5)

**19)** p14, L32-33: I do not wish to take away from the value of the IASI HNO3 measurements, whose dense spatial coverage and long-term record are obviously of great benefit, as this study has shown. But I would ask for a bit more care in the language used here. Although the novel statistical nature of these results is mentioned, I think that some readers could take away from these lines the message that this analysis has revealed the profound influence of PSC formation and denitrification on the HNO3 distribution, when in fact the crucial role of those processes has been known for decades. In truth, it is not obvious to me what additional knowledge about the variability of HNO3 in the polar regions has been gained from this study that had not been demonstrated previously using limb measurements with much coarser horizontal but much greater vertical resolution.

We agree with the reviewer that the results presented in this study have been exposed in previous studies. This work allows confirming well-known processes thanks to the use of a multivariate regression model applied for the first time on $HNO_3$ time series. This sentence was modified: *"The amount of data allows for a thorough monitoring of the processes regulating the $HNO_3$ variability, such as the denitrification processes in the southern polar regions, as well as the seasonal variability in the tropical regions."* (p.16 line 21-23)

**20)** p25, Figure 2: Minor tick marks on the y-axis would be helpful.
Minor ticks on the y-axis have been added on all subplots.

As mentioned earlier, it might be good to remove the sparse measurements during the September-December 2010 interval from this plot as well.

Thank you for this remark. The data for the period in 2010 where too few measurements are available have been removed and the legend of the figure was adapted accordingly.

Why do some of the vertical lines, especially (but not only) in the purple 65-70 eqlat region, appear to be thicker? Is it because temperatures are hovering around the PSC threshold at those times, so the shading is being turned off and on multiple times in quick succession?

Yes indeed, the reason for the impression of thicker lines is due to the fact that the temperatures are oscillating around the 195K threshold. However, the edges of each 'patch' was also darker, in order to identify more clearly the start and end of each period under 195K. After several visual tests, it was decided to remove this darker edge, as it did not seem necessary. This allows limiting the impression of ticker lines (particularly visible in the 65-70 S eqlat band during denitrification). (p. 28)

**21)** p30, Figure 7: Why is there a break in the without-VPSC fit curve in the 70-90S panel in October-November 2014? Such a break does not appear in the similar panel for the with-VPSC fit (or in Figure 6).

We would like to thank the reviewer for that remark. It is actually an error in the plotting that escaped our attention. There is actually no missing data and this was rectified. Note also that there was an error in the figure initially submitted. The time series of IASI, the fit and the residuals (two top rows) were not correctly aligned with the time series of the PSCs. This was also corrected in the revised version of the paper.

**22)** p33, Figure 10 caption: The wording of the caption ("Time evolution of IASI HNO3 (red) and NO2 (green)") implies that the NO2 data are from IASI, but the reference cited is for GOME-2 data. Please clarify.

Indeed, this is a mistake. The $NO_2$ data come from GOME-2, and not from IASI. The caption was rectified and now reads: "*Time evolution of IASI $HNO_3$ (red) and GOME-2 $NO_2$ (green) from 2008 to 2015 for Africa…*" (p. 36)

Minor points of clarification, wording / figure suggestions, and grammar / typo corrections:

**23)** p1, L13: "PSCs" should be defined in the abstract as well as the main body of the paper.

The full name is now used in the abstract, followed by the acronym between parentheses (p.1 line 13-14). The same is done in the text for the first mention (p.1 line 20).

**24)** p1, L23: inexsitent –> non-existent

The change was made (p. 1 line 23).

**25)** p2, L3: "PSCs" was already defined on p1, L20

The full name was replaced by the acronym only (p.2 line 3).

**26)** p2, L8: and further –> followed by

The formulation was modified (p. 2 line 8).

**27)** p2, L11, L14: These acronyms (UARS, MIPAS, ACE-FTS) should probably be spelled out. Also, "AURA" –> "Aura" and "ODIN" –> "Odin" (they are not acronyms, just names)

In order not to have too long sentences and/parentheses, it was decided to spell out the names of the instruments only, and to include them in a footnote. The acronym is kept in the text (p. 2 line 10-16 + footnote).

**28)** p3, L10: bi-daily –> twice daily ("bi-daily" could be interpreted to mean every two days)

The sentence was changed to: "… *consists of measurements taken twice a day (at 9.30 AM and PM, …)"* (p. 3 line 13-14)

**29)** p3, L15: The university name should be spelled out here

The full name is now included, followed by its acronym (p. 3 line 18-19).

**30)** p3, L24: Can 15-20 km really be considered the "low-middle" stratosphere? This seems more like just the lower stratosphere to me.

This was changed to "lower stratosphere" (p.3 line 27).

**31)** p3, L30: higher fractional cloud cover than 25% –> fractional cloud cover higher than 25%

This was modified (p. 3 line 33).

**32)** p4, L19: Further than –> Beyond

The modification was made (p. 4 line 28).

**33)** p4, L22: delete "columns" (some of the previous studies were based on HNO3 profiles, not columns)

The word "columns" was removed (p. 4 line 31).

**34)** p5, L14: delete "itself"

The word "itself" was removed (p. 6 line 4).

**35)** p5, L20: more –> longer

The word "more" was modified to "longer" (p. 6 line 13).

**36)** p5, L26: It would be good to add "Arctic" in front of "winters" and "over a broader area" after "threshold"

These two sentences were adapted: *"… and, to some extent, 2014 Arctic winters. During these three winters, temperatures reached below the 195 K threshold over a broader area and stayed low during a longer period than usual."* (p. 6 line 19-21)

**37)** p5, L35: "polar" –> "potential"

This word was changed (p. 6 line 30).

**38)** p6, L1: it's not clear why only one contour is noted here, when 3 contours of PV are shown in both hemispheres

The reference to the iso-contour was meant only for the last of the cited remarkable features. In order to make this clearer, the sentence was rephrased: *"…, the marked annual cycle at mid to high latitudes and the systematic and the occasional (2011, 2014, 2016) denitrification periods in the high latitudes of the Southern and Northern hemispheres respectively, which are highlighted by the iso-contours of potential vorticity at $\pm10\text{x}10^{-6}$ K.m$^2$.kg$^{-1}$.s$^{-1}$ (dark blue)."* (p. 6 line 27-30)

**39)** p6, L4: EUMETSAT should be in all capital letters (as on p3, L29)

The word was capitalized (p.4  line 1).

**40)** p6, L12: dentrification –> denitrification

The word was corrected (p. 7 line 9).

**41)** p6, L14: What does "more stable" mean in this context? More constant over the season, or more uniform from year to year?? And what is the comparison against - wintertime values in the NH, or summertime values in the SH?

We apologize for the confusion here. The sentence has been rephrased: *"The northern hemisphere high latitudes (top panel) show more interannual variability than in the south, especially during the winter because of the unusual denitrification periods observed in 2011 (purple), 2014 (blue) and 2016 (black) in January (concentrations as low as $2.2x10^{16}$ molec.cm$^{-2}$ in 2016). Contrary to the winter, the summer columns are more uniform from one year to another with values around $2.1x10^{16}$ to $2.8x10^{16}$ molec.cm$^{-2}$."* (p. 7 line 8-11)

**42)** p6, L21: I assume that "Cst" in Eqn (1) is a constant term, but it should be defined

Indeed, it is. This was added in the definition of the various terms (p. 7 line 19).

**43)** p6, L29: Kyrola et al. [2010] seems like an odd reference for such a general statement about the BDC. Wouldn't the Butchart [2014] review paper (already cited elsewhere) be a better choice?

Absolutely, this is quite odd indeed. The reference was modified (p.7 line 26).

**44)** p7, L13: further –> below

The word "further" was replaced by "below" (p. 8 line 11).

**45)** p7, L19: ENSO should also be mentioned here

The ENSO was added: *"… and geophysical proxies for the solar cycle, the QBO, the ENSO phenomenon and for the Arctic (AO) and Antarctic Oscillations (AAO) for the northern and southern hemispheres, respectively."* (p. 8 line 16-18)

**46)** p8, L13: meridonal –> meridional

The word was corrected (p. 9 line 13).

**47)** p8, L23: I assume it is meant that AO/AAO are considered only in the high latitudes of the hemisphere they are related to, since they are both applied in equatorial regions.

That is correct. To avoid any confusion, a precision was added: *"Each index (AO or AAO) is considered only in the hemisphere it is related to, …"* (p. 9 line 23)

**48)** p8, L29: delete "(from 195 K or TNAT for the formation of nitric acid trihydrate particles)"

The parenthesis was deleted (p.9 line 29).

**49)** p8, L30: "PSCs" has already been defined

Only the acronym was used (p.9 line 29).

**50)** p8, L31: delete "either"

The word was removed (p.9 line 30).

**51)** p9, L2: NAT has already been defined

Only the acronym was used (p.10 line 1).

**52)** p9, L3: gaz –> gas

The spelling was corrected (p.10 line 2).

**53)** p9, L13: dynamic –> dynamics

The word was corrected (p. 10 line 12).

**54)** p9, L23: Section 3.2 –> Section 4.2

The number of the section was changed (p. 10 line 22).

**55)** p9, L26: add "bands" after "90S"

The word "bands" was added (p. 10 line 25).

**56)** p9, L29: add "major" after "Most"

The word "major" was added (p. 10 line 28).

**57)** p9, L30: RMSE is used here for the first time but not defined until p11, L31

RMSE was defined here and only the acronym is used further (p. 10 line 29-30).

**58)** p10, L6: denitrifications –> denitrification seasons

The phrasing was modified (p.11 line 6).

**59)** p10, L11: delete "here"

The word "here" was removed (p. 11 line 12).

**60)** p10, L17: better –> improved

The word "better" was replaced by "improved" (p. 11 line 18).

**61)** p10, L23: dynamic –> conditions

The word "dynamic" was replaced by "conditions" (p. 11 line 25).

**62)** p11, L5: The reference to Section 4.4.1 is incorrect (this part of the discussion is itself in Section 4.4.1)

Indeed, we are referring to the first feature that is described in this section. The reference was changed to: *"(see first highlighted feature above)".* (p. 12 line 7)

**63)** p11, L11: most –> much; also add "generally" in front of "between"

The word "most" was replaced by "much". "generally" was added in front of "between" (p. 12 line 13)

**64)** p11, L19: between –> in

The word "between" was replaced by "in". (p. 12 line 22)

**65)** p11, L21: emitted –> produced

The word "emitted" was replaced by "produced" (p. 12 line 23).

**66)** p11, L25: oxydation –> oxidation

The spelling was corrected (p. 12 line 29).

**67)** p11, L34: desertic –> desert

The word was corrected (p. 13 line 3).

**68)** p12, L30: For MEI, "south of Africa" the results are not significant. I think "west of South Africa" would be better here.

Indeed there was some confusion here. It was replaced by the suggested "west of South Africa". (p. 14 line 16)

**69)** p13, L5: Groenland –> Greenland

The spelling was corrected (p. 14 line 21).

**70)** p13, L10: largely –> strongly

The word "largely" was replaced by "strongly" (p. 14 line 26).

**71)** p13, L21: further –> subsequent

The word "further" was replaced by "subsequent" (p. 15 line 9).

**72)** p13, L25: add "proxy" after "VPSC" and "HNO3" in front of "variability"

The word "PSCs" was used instead of "VPSC" (p. 15 line 13).

**73)** p14, L7: reveal –> reflect

The word "reveal" was replaced by "reflect" (p. 15 line 28).

**74)** p14, L14: between –> in

The word "between" was replaced by "in" (p. 16 line 3).

**75)** p14, L23: but still allow improving significantly the model-to-observation agreement –> but accounting for PSCs still significantly improves the model-to-observation agreement

The sentence was adapted as suggested (p. 16 line 11-12).

---

## Author Response (AR2)

Anonymous Referee #2 - Suggestions for revision or reasons for rejection (will be published if the paper is accepted for final publication)

We would like to thank the referee for their thorough review of the revised manuscript. We believe these minor changes improve the manuscript further. All are addressed in the following, in a point-by-point manner, with the referee's comment reproduced in blue first.

1) p5, L20: homogenize --> become homogenized

The word "homogenize" was modified to "become homogenized". (p.5, line 20).

2) p5, L24-34: I have some concerns about the discussion that has been added in these lines. First, I do not believe that it is true that "ozone concentrations return rapidly to usual values almost as soon as PSCs disappear". PSCs formation typically ceases in mid-September, but ozone values remain low well past that date, even in the collar region (and certainly in the vortex core, where the ozone "hole" is in full force through October).

We agree with the referee that the word "rapidly" might be misleading as $O_3$ concentrations stay low in October after the disappearance of PSCs. Our intent was to oppose the relatively rapid increase of $O_3$ columns (which starts mid-October and for which concentrations are back to normal in December) compared to the much slower recovery of $HNO_3$ (which starts only in December and increases more gradually than $O_3$). The sentence in the manuscript was adapted to avoid any confusion: *"However, the recovery of the $HNO_3$ total columns is very slow compared to other species, namely $O_3$, for which concentrations return to usual values within 2 months (i.e. in December) after PSCs have disappeared. In fact, the $HNO_3$ columns stay low well after the September equinox and are only subject to a slow increase starting 2 months later (in early December) and bringing the concentrations back to pre-denitrification levels by May."* (p. 5 lines 23-27).

Second, it is not clear what the word "it" is referring to in L29. It may be that the authors mean that only a few PSCs have not sedimented out of the lower stratosphere and thus remain available, in which case "such that only a few PSCs remain available in the lower stratosphere" would be better than "yielding only small amounts of it".

We thank the referee for bringing up this confusing wording. The sentence has been changed and now reads: *"(1) a significant sedimentation of PSCs towards the lower atmosphere during the winter, such that few PSCs remain available to release $HNO_3$ under warmer temperatures (Lowe and MacKenzie, 2008; Kirner et al., 2011; Khosrawi et al., 2016), …"* (p.5 lines 28-30).

Finally, the action of confined diabatic descent as the vortex spins up at the start of the winter also plays a major role in the increase of HNO3 in March.

We thank the referee for this remark. Mention of the diabatic descent was added: *"The increase observed in March, at the start of the winter, is in turn explainable by a much reduced number of hours of sunlight, implying less photodissociation, as well as by the diabatic descent bringing $HNO_3$-rich air to lower altitudes."* (p.5 lines 34-35).

**3)** p6, L17: I think it would be fairer to say that Arctic lower stratospheric temperatures "frequently" dip below 195 K, rather than that they "sometimes" do. Also, "on broad areas" --> "over broad areas".

The word "sometimes" was modified to "frequently", and the word "on" was replaced by "over'. (p.6 line 18).

**4)** p7, L10: Contrary to --> In contrast to

The word "Contrary to" was modified to "In contrast to". (p.7 line 11).

**5)** p13, L5-6: I had missed the significance of these lines in reading the previous manuscript. Just to make sure that the point is totally clear to all readers, the authors might change the wording here to "Regions of low clouds or those characterized by emissivity features that are sharp (e.g., deserts) or seasonally varying (e.g., ice shelves) are known to cause problems for the retrieval of HNO3 using the IASI spectra". Then on p14, L10, they could add "the ice shelves of" between "above" and "western Antarctica".

We thank the referee for this comment. We agree that this wording makes things clearer for all readers, and the sentence was thus changed accordingly. (p. 13 lines 5-7).

Mention of the ice shelves was also added in the following section when describing the distribution of the fitted proxies. (p. 14 line 9).

**6)** p14, L5-7: I found the text added here confusing. First the influence of SF on HNO3 is described. Then it is stated that previous studies have shown that SF contributions to O3 variability are largest in the lower stratosphere. This is followed by the sentence: "Our results for the mid to high latitudes suggest opposite behaviour for HNO3 (as was also reported for O3 by Wespes et al. (2017))." It is not clear what "opposite" means here; since the immediately preceding sentence talks about the O3 response being largest in the lower stratosphere, as written this seems to imply that that is not the case for HNO3, but I think the authors meant instead that the geographic distribution of sensitivity to SF of HNO3 is not the same as that of O3. Moreover, the parenthetical saying that similar results were reported for O3 by Wespes et al. (2017) appears to contradict the first part of the sentence mentioning "opposite" behavior. This may be because the latter paper focuses on tropospheric, not stratospheric, ozone. In any case, this part of the discussion needs to be clarified.

We would like to thank the referee for pointing out this section. Indeed it needs to be clarified. What we meant is that we obtain different results than in other studies. The reference to Wespes et al. (2017) was initially meant to show that different results have also been obtained in other studies, but since the pattern is not that similar to ours and it is a study of the tropospheric $O_3$, it was chosen to remove that reference altogether. The sentence was changed in order to make things clearer: *"The influence of the solar flux is positive in the northern polar latitudes and in the tropical and southern mid-latitudes. It is close to zero or negative elsewhere. While previous studies showed a positive signal globally in the low stratosphere for the response of $O_3$ to the solar cycle (Hood, 1997; Soukharev and Hood, 2006), our results for the mid to high northern latitudes suggest opposite behaviour (negative signal) for $HNO_3$. However, the positive contribution of the solar cycle on the $HNO_3$ variation in the tropical and southern mid-latitude stratosphere is in line with the $O_3$ response previously reported (Soukharev and Hood, 2006; McCormack et al., 2007; Frossard et al., 2013; Maycock et al., 2016)."* (p. 14 lines 2-9).

**7)** p15, L2: It would be clearer to say "Arctic" instead of "polar" here.

The word "polar" was replaced by "Arctic". (p. 15 line 1).

**8)** Figure 2: I do think that it was a good idea to remove from this plot the period in September to December 2010 when the amount of data distributed by EUMETSAT was abnormally small. However, I noticed that the results in all four EqL bands changed substantially from those in the original manuscript throughout all of 2010 (even in the January to April 2010 timeframe, well before the data gap), particularly in the south. I was not expecting such a large change to the curves outside of the interval directed affected by the data shortage when those dates were removed from the analysis.

We would like to thank the referee for this remark. This is a mistake that escaped our attention, and this is obviously wrong. We updated the figure and it now shows no difference with the previous version, except for the removed data from September to December (Figure 2, page 28).

[revised manuscript text omitted]